# MaxSup: Overcoming Representation Collapse in Label Smoothing

**Yuxuan Zhou**$^{\alpha,\beta*}$   **Heng Li**$^{\gamma,*\dagger}$   **Zhi-Qi Cheng**$^{\gamma,\epsilon\ddagger}$   **Xudong Yan**$^{\gamma,\dagger}$   **Yifei Dong**$^{\gamma}$
**Mario Fritz**$^{\beta}$   **Margret Keuper**$^{\alpha,\delta}$

$^{\alpha}$ University of Mannheim    $^{\gamma}$ University of Washington    $^{\epsilon}$ Meta AI
$^{\beta}$ CISPA Helmholtz Center for Information Security    $^{\delta}$ Max Planck Institute for Informatics

## Abstract

Label Smoothing (LS) is widely adopted to reduce overconfidence in neural network predictions and improve generalization. Despite these benefits, recent studies reveal two critical issues with LS. First, LS induces overconfidence in misclassified samples. Second, it compacts feature representations into overly tight clusters, diluting intra-class diversity, although the precise cause of this phenomenon remained elusive. In this paper, we analytically decompose the LS-induced loss, exposing two key terms: *(i) a regularization term* that dampens overconfidence only when the prediction is correct, and *(ii) an error-amplification term* that arises under misclassifications. This latter term compels the network to reinforce incorrect predictions with undue certainty, exacerbating representation collapse. To address these shortcomings, we propose Max Suppression (MaxSup), which applies uniform regularization to both correct and incorrect predictions by penalizing the top-1 logit rather than the ground-truth logit. Through extensive feature-space analyses, we show that MaxSup restores intra-class variation and sharpens inter-class boundaries. Experiments on large-scale image classification and multiple downstream tasks confirm that MaxSup is a more robust alternative to LS. [4]

## 1 Introduction

Multi-class classification [19, 26] typically relies on one-hot labels, which implicitly treat different classes as mutually orthogonal. In practice, however, classes often share low-level features [31, 44] or exhibit high-level semantic similarities [3, 24, 42], rendering the one-hot assumption overly restrictive. Such a mismatch can yield over-confident classifiers and ultimately degrade generalization [9].

To moderate overconfidence, Szegedy et al. [34] introduced Label Smoothing (LS), which combines a uniform distribution with the hard ground-truth label, thereby reducing the model's certainty in the primary class. LS has since become prevalent in image recognition [10, 22, 36, 47] and neural machine translation [1, 6], often boosting accuracy and calibration [23]. Yet subsequent work indicates that LS can overly compress features into tight clusters [15, 28, 41], hindering intra-class variability and transferability [5]. In parallel, Zhu et al. [48] found that LS paradoxically fosters overconfidence in misclassified samples, though the precise mechanism behind this remains uncertain.

In this paper, we reveal that LS's training objective inherently contains an error amplification term. This term pushes the network to reinforce incorrect predictions with exaggerated certainty, yielding

---

*Equal contribution.

$^{\dagger}$Internship at University of Washington.

$^{\ddagger}$Corresponding author. Assistant Professor, UW Tacoma School of Engineering and Technology.

[4]`https://github.com/ZhouYuxuanYX/Maximum-Suppression-Regularization`

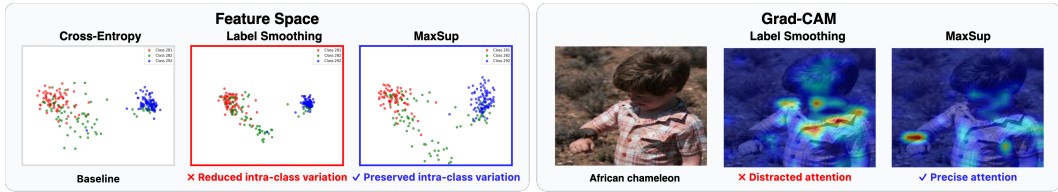

Figure 1: Comparison of Label Smoothing (LS) and MaxSup. **Left:** MaxSup mitigates the intra-class compression induced by LS while preserving inter-class separability. **Right:** Grad-CAM visualizations show that MaxSup more effectively highlights class-discriminative regions than LS.

highly confident misclassifications and further compressing feature clusters (Section 3.1, Table 1). Building on Zhu et al. [48], we characterize "overconfidence" in terms of the model's top-1 prediction, rather than through conventional calibration metrics. Through our analysis, we further show that punishing the ground-truth logit during misclassification reduces intra-class variation (Table 2), a phenomenon corroborated by Grad-CAM visualizations (Figure 2).

To overcome these shortcomings, we introduce Max Suppression (MaxSup), a method that retains the beneficial regularization effect of LS while eliminating its error amplification. Rather than penalizing the ground-truth logit, MaxSup focuses on the model's top-1 logit, ensuring a consistent regularization signal regardless of whether the current prediction is correct or misclassified. By preserving the ground-truth logit in misclassifications, MaxSup sustains richer intra-class variability and sharpens inter-class boundaries. As visualized in Figure 1, this approach mitigates the feature collapse and attention drift often induced by LS, ultimately leading to more robust representations. Through comprehensive experiments in both image classification (Section 4.2) and semantic segmentation (Section 4.3), we show that MaxSup not only alleviates severe intra-class collapse but also consistently boosts top-1 accuracy and robustly enhances downstream transfer performance (Section 4.1).

Our contributions are summarized as follows:

- We perform a logit-level analysis of Label Smoothing, revealing how the error amplification term inflates misclassification confidence and compresses features.
- We propose Max Suppression (MaxSup), removing detrimental error amplification while preserving LS's beneficial regularization. As shown in extensive ablations, MaxSup alleviates intra-class collapse and yields consistent accuracy gains.
- We demonstrate superior performance across tasks and architectures, including ResNet, MobileNetV2, and DeiT-S, where MaxSup significantly boosts accuracy on ImageNet and consistently delivers stronger representations for downstream tasks such as semantic segmentation and robust transfer learning.

## 2 Related Work

We first outline mainstream regularization techniques in deep learning, then survey recent advances in Label Smoothing (LS), and finally clarify how our MaxSup diverges from prior variants.

### 2.1 Regularization

Regularization techniques aim to improve the generalization of deep neural networks by constraining model complexity. Classical methods like $\ell_2$ [18] and $\ell_1$ [49] impose direct penalties on large or sparse weights, while Dropout [32] randomly deactivates neurons to discourage over-adaptation. In the realm of loss-based strategies, Label Smoothing (LS) [34] redistributes a fraction of the label probability mass away from the ground-truth class, thereby improving accuracy and calibration [23]. Variants such as Online Label Smoothing (OLS) [45] and Zipf Label Smoothing (Zipf-LS) [21] refine LS by dynamically adjusting the smoothed labels based on a model's evolving predictions. However, they do not fully address the fundamental issue that emerges when the ground-truth logit is not the highest one (see Section 3.1, Table 1). Other loss-based regularizers focus on alternative aspects of the predictive distribution. Confidence Penalty [25] penalizes the model's confidence directly, while Logit Penalty [4] minimizes the global $\ell_2$-norm of logits, a technique reported to enhance class separation [15]. Despite these benefits, Logit Penalty can inadvertently shrink intra-class variation,

thereby hampering transfer learning (see Section 4.1). Unlike the aforementioned methods, MaxSup enforces regularization by penalizing only the top-1 logit $z_{max}$ rather than the ground-truth logit $z_{gt}$. In LS-based approaches, suppressing $z_{gt}$ for misclassified samples can worsen errors, whereas MaxSup applies a uniform penalty regardless of whether the model's prediction is correct. Consequently, MaxSup avoids the error amplification effect, retains richer intra-class diversity (see Table 2), and achieves robust transfer performance across diverse datasets and model families (see Table 3).

## 2.2 Studies on Label Smoothing

Label Smoothing has also been studied extensively under knowledge distillation. For instance, Yuan et al. [43] observed that LS can approximate the effect of a teacher–student framework, while Shen et al. [30] investigated its role in such pipelines more systematically. Additionally, Chandrasegaran et al. [2] demonstrated that a low-temperature, LS-trained teacher can notably improve distillation outcomes. Concurrently, Kornblith et al. [15] showed that LS tightens intra-class clusters in the feature space, diminishing transfer performance. From a Neural Collapse perspective [46, 8], LS nudges the model toward rigid feature clusters, as evidenced by the reduced feature variability measured in Xu and Liu [41]. Our goal is to overcome LS's inherent error amplification effect. Rather than adjusting how the smoothed label distribution is constructed (as in OLS or Zipf-LS), MaxSup directly penalizes the highest logit $z_{max}$. This design ensures consistent regularization even if $z_{gt}$ is not the top logit, thereby avoiding the degradation in performance typical of misclassified samples under LS (see Section 3.2). Moreover, MaxSup integrates seamlessly into standard training pipelines, introducing negligible computational overhead beyond substituting the LS term.

## 3 Max Suppression Regularization (MaxSup)

We first partition the training objective into two components: the standard Cross-Entropy (CE) loss and a regularization term introduced by Label Smoothing (LS). By expressing LS in terms of logits (Theorem 3.3), we isolate two key factors: a regularization term that controls overconfidence and an error amplification term that enlarges the gap between the ground-truth logit $z_{gt}$ and any higher logits (Theorem 3.4, Equation (5)), ultimately degrading performance. To address these issues, we propose Max Suppression Regularization (MaxSup), which applies the penalty to the largest logit $z_{max}$ rather than $z_{gt}$ (Equation (8), Section 3.2). This shift delivers consistent regularization for both correct and incorrect predictions, preserves intra-class variation, and bolsters inter-class separability. Consequently, MaxSup mitigates the representation collapse found in LS, attains superior ImageNet-1K accuracy (Table 1), and improves transferability (Table 2, Table 3). The following sections elaborate on MaxSup's formulation and integration into the training pipeline.

## 3.1 Revisiting Label Smoothing

Label Smoothing (LS) is a regularization technique designed to reduce overconfidence by softening the target distribution. Rather than assigning probability 1 to the ground-truth class and 0 to all others, LS redistributes a fraction $\alpha$ of the probability uniformly across all classes:

**Definition 3.1.** For a standard classification task with $K$ classes, Label Smoothing (LS) converts a one-hot label $\mathbf{y} \in \mathbb{R}^K$ into a softened target label $\mathbf{s} \in \mathbb{R}^K$:

$$s_k = (1 - \alpha)y_k + \frac{\alpha}{K}, \tag{1}$$

where $y_k = \mathbb{1}_{\{k=gt\}}$ denotes the ground-truth class. The smoothing factor $\alpha \in [0, 1]$ reduces the confidence assigned to the ground-truth class and distributes $\frac{\alpha}{K}$ to other classes uniformly, thereby mitigating overfitting, enhancing robustness, and promoting better generalization.

To clarify the effect of LS on model training, we first decompose the Cross-Entropy (CE) loss into a standard CE term and an additional LS-induced regularization term:

**Lemma 3.2.** *Decomposition of Cross-Entropy Loss with Soft Labels.*

$$H(\mathbf{s}, \mathbf{q}) = H(\mathbf{y}, \mathbf{q}) + L_{LS}, \tag{2}$$

*where*

$$L_{LS} = \alpha \left( H\left(\tfrac{1}{K}, \mathbf{q}\right) - H(\mathbf{y}, \mathbf{q}) \right). \tag{3}$$

Where, $\mathbf{q}$ is the predicted probability vector, $H(\cdot)$ denotes the Cross-Entropy, and $\frac{1}{K}$ is the uniform distribution introduced by LS. This shows that LS adds a regularization term, $L_{LS}$, which smooths the output distribution and helps to reduce overfitting. (See Section A for a formal proof.)

Building on Theorem 3.2, we next explicitly express $L_{LS}$ at the logit level for further analysis.

**Theorem 3.3.** *Logit-Level Formulation of Label Smoothing Loss.*

$$L_{LS} = \alpha\Big(z_{gt} - \frac{1}{K}\sum_{k=1}^{K} z_k\Big), \tag{4}$$

where $z_{gt}$ is the logit corresponding to the ground-truth class, and $\frac{1}{K}\sum_{k=1}^{K} z_k$ is the average logit. Thus, LS penalizes the gap between $z_{gt}$ and the average logit, encouraging a more balanced output distribution and reducing overconfidence. (See Section B for the proof.)

The behavior of $L_{LS}$ differs depending on whether $z_{gt}$ is already the maximum logit. Specifically, depending on whether the prediction is correct ($z_{gt} = z_{max}$) or incorrect ($z_{gt} \neq z_{max}$), we can decompose $L_{LS}$ into two parts:

**Corollary 3.4.** *Decomposition of Label Smoothing Loss.*

$$L_{LS} = \underbrace{\frac{\alpha}{K}\sum_{z_m < z_{gt}}\Big(z_{gt} - z_m\Big)}_{Regularization} + \underbrace{\frac{\alpha}{K}\sum_{z_n > z_{gt}}\Big(z_{gt} - z_n\Big)}_{Error\ amplification}, \tag{5}$$

where $M$ and $N$ are the numbers of logits below and above $z_{gt}$, respectively ($M + N = K - 1$). Note that the error amplification term vanishes when $z_{gt} = z_{max}$.

1. **Regularization**: Penalizes the gap between $z_{gt}$ and any smaller logits, thereby moderating overconfidence.

2. **Error amplification**: Penalizes the gap between $z_{gt}$ and larger logits, inadvertently increasing overconfidence in incorrect predictions.

Although LS aims to combat overfitting by reducing prediction confidence, its error amplification component can be detrimental for misclassified samples, as it widens the gap between the ground-truth logit $z_{gt}$ and the incorrect top logit. Concretely:

1. **Correct Predictions** ($z_{gt} = z_{max}$): The error amplification term is zero, and the regularization term effectively reduces overconfidence by shrinking the gap between $z_{gt}$ and any smaller logits.

2. **Incorrect Predictions** ($z_{gt} \neq z_{max}$): LS introduces two potential issues:
   - **Error amplification**: Increases the gap between $z_{gt}$ and larger logits, reinforcing overconfidence in incorrect predictions.
   - **Inconsistent Regularization**: The regularization term lowers $z_{gt}$ yet does not penalize $z_{max}$, which further impairs learning.

These issues with LS on misclassified samples have also been systematically observed in prior work [39]. By precisely disentangling these two components (regularization vs. error amplification), we can design a more targeted and effective solution.

**Ablation Study on LS Components.** To isolate the effects of each component in LS, we carefully perform a detailed and systematic ablation study on ImageNet-1K using a DeiT-Small model [36] without Mixup or CutMix. As indicated in Table 1, the performance gains from LS stem solely from the regularization term, whereas the error amplification term degrades accuracy. In contrast, our MaxSup omits the error amplification component and leverages only the beneficial regularization, thereby boosting accuracy beyond that of standard LS. Specifically, Table 1 shows that LS's overall improvement can be attributed exclusively to its regularization contribution; the error amplification term consistently reduces accuracy (e.g., to 73.63% or 73.69%). Disabling only the error amplification while retaining the regularization yields a slight but measurable improvement (75.98% vs. 75.91%). By fully removing error amplification and faithfully preserving the helpful aspects of LS, our MaxSup achieves 76.12% accuracy, clearly and consistently outperforming LS. This result underscores that MaxSup directly tackles LS's fundamental shortcoming by maintaining a consistent and meaningful regularization signal—even when the top-1 prediction is incorrect.

Table 1: Ablation on LS components using DeiT-Small on ImageNet-1K (without CutMix or Mixup). "Regularization" denotes penalizing logits smaller than $z_{gt}$; "error amplification" penalizes logits larger than $z_{gt}$. MaxSup removes error amplification while retaining regularization.

| Method | Formulation | Accuracy |
|---|---|---|
| Baseline | – | 74.21 |
| + Label Smoothing | $\frac{\alpha}{K}\sum_{z_m<z_{gt}}(z_{gt}-z_m)$ $+\frac{\alpha}{K}\sum_{z_n>z_{gt}}(z_{gt}-z_n)$ | 75.91 |
| + Regularization | $\frac{\alpha}{M}\sum_{z_m<z_{gt}}(z_{gt}-z_m)$ | 75.98 |
| + error amplification | $\frac{\alpha}{N}\sum_{z_n>z_{gt}}(z_{gt}-z_n)$ | 73.63 |
| + error amplification | $\alpha\left(z_{gt}-z_{max}\right)$ | 73.69 |
| + MaxSup | $\alpha\left(z_{max}-\frac{1}{K}\sum_{k=1}^{K}z_k\right)$ | 76.12 |

## 3.2 Max Suppression Regularization

Building on our analysis in Section 3.1, we find that Label Smoothing (LS) not only impacts correctly classified samples but also influences misclassifications in unintended and harmful ways. Specifically, LS suffers from two main limitations: inconsistent regularization and error amplification. As illustrated in Table 1, LS penalizes the ground-truth logit $z_{gt}$ even in misclassified examples, needlessly widening the gap between $z_{gt}$ and the erroneous top-1 logit. To resolve these critical shortcomings, we propose Max Suppression Regularization (MaxSup), which explicitly penalizes the largest logit $z_{max}$ rather than $z_{gt}$. This key design choice ensures uniform regularization across both correct and misclassified samples, effectively eliminating the error-amplification issue in LS (Table 1) and preserving the ground-truth logit's integrity for more stable, robust learning.

**Definition 3.5. Max Suppression Regularization**

We define the Cross-Entropy loss with MaxSup as follows:

$$\underbrace{H(\mathbf{s},\mathbf{q})}_{\text{CE with Soft Labels}} = \underbrace{H(\mathbf{y},\mathbf{q})}_{\text{CE with Hard Labels}} + \underbrace{L_{MaxSup}}_{\text{Max Suppression Loss}}, \tag{6}$$

where

$$L_{MaxSup} = \alpha\left(H\left(\tfrac{1}{K},\mathbf{q}\right) - H(\mathbf{y}',\mathbf{q})\right), \tag{7}$$

and

$$y'_k = \mathbb{1}_{\left\{k=\arg\max(\mathbf{q})\right\}},$$

so that $y'_k = 1$ identifies the model's top-1 prediction and $y'_k = 0$ otherwise. Here, $H\left(\tfrac{1}{K},\mathbf{q}\right)$ encourages a uniform output distribution to mitigate overconfidence, while $H(\mathbf{y}',\mathbf{q})$ penalizes the current top-1 logit. By shifting the penalty from $z_{gt}$ (the ground-truth logit) to $z_{max}$ (the highest logit), MaxSup avoids unduly suppressing $z_{gt}$ when the model misclassifies, thus overcoming Label Smoothing's principal shortcoming.

**Logit-Level Formulation of MaxSup.** Building on the logit-level perspective introduced for LS in Section 3.1, we can express $L_{MaxSup}$ as:

$$L_{MaxSup} = \alpha\left(z_{max} - \frac{1}{K}\sum_{k=1}^{K}z_k\right), \tag{8}$$

where $z_{max} = \max_k\{z_k\}$ is the largest (top-1) logit, and $\frac{1}{K}\sum_{k=1}^{K}z_k$ is the mean logit. Unlike LS, which penalizes the ground-truth logit $z_{gt}$ and may worsen errors in misclassified samples, MaxSup shifts the highest logit uniformly, thus providing consistent regularization for both correct and incorrect predictions. As shown in Table 1, this approach eliminates LS's error-amplification issue while preserving the intended overconfidence suppression.

**Comparison with Label Smoothing.** MaxSup fundamentally differs from LS in handling correct and incorrect predictions. When $z_{gt} = z_{max}$, both LS and MaxSup similarly reduce overconfidence. However, when $z_{gt} \neq z_{max}$, LS shrinks $z_{gt}$, widening the gap with the incorrect logit, whereas

MaxSup penalizes $z_{max}$, preserving $z_{gt}$ from undue suppression. As illustrated in Figure 2, this helps the model recover from mistakes more effectively and avoid reinforcing incorrect predictions.

**Gradient Analysis.** To understand MaxSup's optimization dynamics, we compute its gradients with respect to each logit $z_k$. Specifically,

$$\frac{\partial L_{MaxSup}}{\partial z_k} = \begin{cases} \alpha\left(1 - \frac{1}{K}\right), & \text{if } k = \arg\max(\mathbf{q}), \\ -\frac{\alpha}{K}, & \text{otherwise.} \end{cases} \tag{9}$$

Thus, the top-1 logit $z_{max}$ is reduced by $\alpha\left(1 - \frac{1}{K}\right)$, while all other logits slightly increase by $\frac{\alpha}{K}$. In misclassified cases, the ground-truth logit $z_{gt}$ is spared from penalization, avoiding the error-amplification issue seen in LS. For completeness, Appendix A provides the full gradient derivation. While [39] conducted a related gradient analysis of the training loss, it focuses specifically on the setting of selective classification, and examines a posthoc logit normalization technique to mitigate confidence calibration issues. However, this approach addresses only the overconfidence problem of label smoothing (LS), without tackling representation collapse. Moreover, our work presents a logit-level reformulation of LS that provides a deeper theoretical understanding of why LS amplifies errors.

**Behavior Across Different Samples.** MaxSup applies a dynamic penalty based on the model's current predictions. For high-confidence, correctly classified examples, it behaves similarly to LS by reducing overconfidence, effectively mitigating overfitting. In contrast, for misclassified or uncertain samples, MaxSup aggressively suppresses the incorrect top-1 logit, further safeguarding the ground-truth logit $z_{gt}$. This selective strategy preserves a faithful and reliable representation of the true class while actively discouraging error propagation. As shown in Section 4.2 and Table 5, this promotes more robust decision boundaries and leads to stronger generalization.

**Theoretical Insights and Practical Benefits.** MaxSup provides both theoretical and practical advantages over LS. Whereas LS applies a uniform penalty to the ground-truth logit regardless of correctness, MaxSup penalizes only the most confident logit $z_{max}$. This dynamic adjustment robustly prevents error accumulation in misclassifications, ensuring more stable convergence. As a result, MaxSup generalizes better and achieves strong performance on challenging datasets. Moreover, as shown in Section 4.1, MaxSup preserves greater intra-class diversity, substantially improving transfer learning (Table 3) and yielding more interpretable activation maps (Figure 2).

## 4 Experiments

We begin by examining how MaxSup improves feature representations, then evaluate it on large-scale image classification and semantic segmentation tasks. Finally, we visualize class activation maps to illustrate the practical benefits of MaxSup.

### 4.1 Analysis of MaxSup's Learning Benefits

Having established how MaxSup addresses Label Smoothing's (LS) principal shortcomings (Section 3.1), we now demonstrate its impact on inter-class separability and intra-class variation—two properties essential for accurate classification and effective transfer learning.

### 4.1.1 Intra-Class Variation and Transferability

As noted in Section 3.1, Label Smoothing (LS) primarily curbs overconfidence for correctly classified samples but inadvertently triggers error amplification in misclassifications. This uneven penalization can overly compress intra-class feature representations. By contrast, MaxSup uniformly penalizes the top-1 logit, whether the prediction is correct or incorrect, thereby eliminating LS's error-amplification effect and preserving finer distinctions within each class.

Table 2: Feature quality of ResNet-50 on ImageNet-1K.

| Method | $\bar{d}_{\text{within}}\uparrow$ | | $R^2\uparrow$ | |
|---|---|---|---|---|
| | Train | Val | Train | Val |
| Baseline | 0.311 | 0.331 | 0.403 | 0.445 |
| LS | 0.263 | 0.254 | 0.469 | 0.461 |
| OLS | 0.271 | 0.282 | 0.594 | 0.571 |
| Zipf-LS | 0.261 | 0.293 | 0.552 | 0.479 |
| MaxSup | **0.293** | 0.300 | 0.519 | 0.497 |
| Logit Penalty | 0.284 | **0.314** | 0.645 | 0.602 |

Table 2 compares intra-class variation ($\bar{d}_{\text{within}}$) and inter-class separability ($R^2$) [15] for ResNet-50 trained on ImageNet-1K. Although all investigated regularizers decrease $\bar{d}_{\text{within}}$ relative to a baseline, MaxSup yields the smallest reduction, indicating a stronger retention of subtle within-class diversity—widely associated with enhanced generalization and improved transfer performance.

These benefits are further underscored by the linear-probe transfer accuracy on CIFAR-10 (Table 3). While LS and Logit Penalty each boost ImageNet accuracy, both degrade transfer accuracy, likely by suppressing informative and transferable features. By contrast, MaxSup preserves near-baseline performance, implying that it maintains rich discriminative information crucial for downstream tasks. For extended evaluations on diverse datasets, see Table 12 in the appendix.

### 4.1.2 Connection to Logit Penalty

As detailed in Section 3, both Label Smoothing (LS) variants and MaxSup impose penalties directly at the logit level, aligning with the perspective that various regularizers influence a model's representational capacity via distinct logit constraints [15]. Within this family of techniques, Logit Penalty and MaxSup both address the maximum logit, yet diverge fundamentally in their specific methods of regularization.

Table 3: Linear-probe transfer accuracy on CIFAR-10 (higher is better).

| Method | Acc. |
|---|---|
| Baseline | 0.814 |
| Label Smoothing | 0.746 |
| Logit Penalty | 0.724 |
| MaxSup | **0.810** |

Logit Penalty minimizes the $\ell_2$-norm of the entire logit vector, inducing a global contraction that can improve class separation but also reduce intra-class diversity, potentially hindering downstream transfer. By contrast, MaxSup focuses exclusively on the top-1 logit, gently nudging it closer to the mean logit. Because only the highest-confidence prediction is penalized, MaxSup avoids the uniform shrinkage observed in Logit Penalty, preserving richer intra-class variation—a property essential for robust transfer. Further insights into this behavior can be found in Section L, where logit-value histograms illustrate how each method affects the logit distribution.

## 4.2 Evaluation on ImageNet Classification

Next, we compare MaxSup to standard Label Smoothing (LS) and various LS extensions on the large-scale ImageNet-1K dataset.

### 4.2.1 Experiment Setup

**Model Training Configurations.**We evaluate both convolutional (ResNet[10], MobileNetV2 [27]) and transformer (DeiT-Small [36]) architectures on ImageNet [17]. For the **ResNet Series**, we train for 200 epochs using stochastic gradient descent (SGD) with momentum0.9, weight decay of $1 \times 10^{-4}$, and a batch size of 2048. The initial learning rate is 0.85 and is annealed via a cosine schedule.[5] We also test ResNet variants on CIFAR-100 with a conventional setup: an initial learning rate of 0.1 (reduced fivefold at epochs 60, 120, and 160), training for 200 epochs with batch size 128 and weight decay $5 \times 10^{-4}$. For **DeiT-Small**, we use the official codebase [36], training from scratch without knowledge distillation to isolate MaxSup's contribution. CutMix and Mixup are disabled to ensure the model optimization objective remains unchanged.

**Hyperparameters for Compared Methods.**We compare Max Suppression Regularization against a range of LS extensions, including Zipf Label Smoothing[21] and Online Label Smoothing [45]. Where official implementations exist, we adopt them directly; otherwise, we follow the methodological details provided in each respective paper. Except for any method-specific hyperparameters, all other core training settings remain identical to the baselines. Furthermore, both MaxSup and standard LS employ a linearly increasing $\alpha$-scheduler for improved training stability (see Section F). This ensures a fair comparison under consistent and reproducible training protocols.

### 4.2.2 Experiment Results

**ConvNet Comparison.** Table4 shows results for MaxSup alongside various label-smoothing and self-distillation methods on both ImageNet and CIFAR-100 benchmarks. Across all convolutional

---

[5]Additional training hyperparameters follow the FFCV scripts at `https://github.com/libffcv/ffcv`. See Section E for further details.

architectures tested, MaxSup consistently delivers the highest top-1 accuracy among label-smoothing approaches. By contrast, OLS [45] and Zipf-LS [21] exhibit less stable gains, suggesting their effectiveness may heavily hinge on specific training protocols.

To reproduce OLS and Zipf-LS, we apply the authors' official codebases and hyperparameters but do not replicate their complete training recipes (e.g., OLS trains for 250 epochs with a step-scheduled learning rate of 0.1, and Zipf-LS uses 100 epochs with distinct hyperparameters). Even under these modified settings, MaxSup remains robust, highlighting its effectiveness across a variety of training schedules—unlike the more schedule-sensitive improvements noted for OLS and Zipf-LS.

Table 4: Performance comparison of classical convolutional networks on ImageNet and CIFAR-100. All results are shown as "mean ± std" (percentage). **Bold** highlights the best performance; underlined marks the second best. (Methods labeled with $^*$ indicate code adapted from official repositories; see the text for additional details.)

| Method | ImageNet | | | | CIFAR-100 | | | |
|---|---|---|---|---|---|---|---|---|
| | ResNet-18 | ResNet-50 | ResNet-101 | MobileNetV2 | ResNet-18 | ResNet-50 | ResNet-101 | MobileNetV2 |
| Baseline | 69.09±0.12 | 76.41±0.10 | 75.96±0.18 | 71.40±0.12 | 76.16±0.18 | 78.69±0.16 | 79.11±0.21 | 68.06±0.06 |
| Label Smoothing | 69.54±0.15 | 76.91±0.11 | 77.37±0.15 | 71.61±0.09 | 77.05±0.17 | 78.88±0.13 | 79.19±0.25 | 69.65±0.08 |
| Zipf-LS$^*$ | 69.31±0.12 | 76.73±0.17 | 76.91±0.11 | 71.16±0.15 | 76.21±0.12 | 78.75±0.21 | 79.15±0.18 | 69.39±0.08 |
| OLS$^*$ | 69.45±0.15 | 77.23±0.21 | 77.71±0.17 | 71.63±0.11 | 77.33±0.15 | 78.79±0.12 | 79.25±0.15 | 68.91±0.11 |
| MaxSup | **69.96±0.13** | **77.69±0.07** | **78.18±0.12** | **72.08±0.17** | **77.82±0.15** | **79.15±0.13** | **79.41±0.19** | **69.88±0.07** |
| Logit Penalty | 68.48±0.10 | 76.73±0.10 | 77.20±0.15 | 71.13±0.10 | 76.41±0.15 | 78.90±0.16 | 78.89±0.21 | 69.46±0.08 |

**DeiT Comparison.** Table 5 summarizes performance for DeiT-Small on ImageNet across various regularization strategies. Notably, MaxSup attains a top-1 accuracy of 76.49%, surpassing standard Label Smoothing by 0.41%. In contrast, LS variants such as Zipf-LS and OLS offer only minor gains over LS, implying that their heavy reliance on data augmentation may limit their applicability to vision transformers. By outperforming both LS and its variants without additional data manip-

Table 5: DeiT-Small top-1 accuracy (%), reported as mean ± standard deviation. Values in parentheses indicate absolute improvements over the baseline.

| Method | Mean | Std |
|---|---|---|
| Baseline | 74.39 | 0.19 |
| Label Smoothing | 76.08 (+1.69) | 0.16 |
| Zipf-LS | 75.89 (+1.50) | 0.26 |
| OLS | 76.16 (+1.77) | 0.18 |
| MaxSup | **76.49 (+2.10)** | **0.12** |

ulations, MaxSup demonstrates robust feature enhancement. These findings underscore MaxSup's adaptability to different architectures and emphasize its utility in scenarios where conventional label-smoothing methods yield limited benefits.

**Fine-Grained Classification.** Beyond large-scale benchmarks like ImageNet, we further evaluate MaxSup on two fine-grained visual recognition tasks: CUB-200-2011 [37] and Stanford Cars [16]. These datasets pose unique challenges due to subtle inter-class differences, which often expose the limitations of standard regularization approaches. As shown in Table 6, MaxSup achieves the best performance across both datasets, surpassing LS and

Table 6: Classification on CUB and Cars Datasets.

| Method | CUB[37] | Cars[16] |
|---|---|---|
| Baseline | 80.88 | 90.27 |
| LS | 81.96 | 91.64 |
| OLS | 82.33 | 91.96 |
| Zipf-LS | 81.40 | 90.99 |
| MaxSup | **82.53** | **92.25** |

its recent variants. This demonstrates that MaxSup encourages the model to learn more discriminative and semantically rich representations that better capture fine-grained attributes, such as textures and part-level details. The consistent improvements on these benchmarks further validate MaxSup's capacity to generalize across different visual domains and its potential to enhance robustness in recognition scenarios where nuanced feature understanding is critical.

**Long-Tailed Classification.** To assess the effectiveness of MaxSup under data imbalance, we performed experiments on the CIFAR-10-LT dataset with imbalance ratios of 50 and 100, following the experimental settings described in [35]. The corresponding results are summarized in Table 7. The evaluation compares three setups: Focal Loss, Focal Loss + LS, and Focal Loss + MaxSup. Across all imbalance ratios and splits (val/test), MaxSup consistently outperforms both the baseline and LS in overall accuracy, which jointly reflects the many-shot, medium-shot, and low-shot (minor class) performance. For example, at an imbalance ratio of 50 on the test split, MaxSup achieves 81.4% accuracy, outperforming Focal Loss (76.8%) by 4.6 percentage points, and LS (80.5%) by

Table 7: Comparison of classification performance (%) across imbalance levels for different loss strategies (Focal Loss vs Label Smoothing (LS) vs MaxSup) on the long-tailed CIFAR-10 dataset using Resnet-32. Best performances are in bold.

| Dataset | Split | Imbalance Ratio | Method | Overall | Many | Medium | Low |
|---|---|---|---|---|---|---|---|
| LT CIFAR-10 | val | 50 | Focal Loss | 77.4 | 76.0 | 89.7 | 0.0 |
| | | | Label Smoothing | 81.2 | 81.6 | 77.0 | 0.0 |
| | | | MaxSup | **82.1** | 82.5 | 78.1 | 0.0 |
| LT CIFAR-10 | test | 50 | Focal Loss | 76.8 | 75.3 | 90.4 | 0.0 |
| | | | Label Smoothing | 80.5 | 81.1 | 75.4 | 0.0 |
| | | | MaxSup | **81.4** | 82.3 | 73.4 | 0.0 |
| LT CIFAR-10 | val | 100 | Focal Loss | 75.1 | 71.8 | 88.3 | 0.0 |
| | | | Label Smoothing | 76.6 | 80.6 | 60.7 | 0.0 |
| | | | MaxSup | **77.1** | 80.1 | 65.1 | 0.0 |
| LT CIFAR-10 | test | 100 | Focal Loss | 74.7 | 71.6 | 87.2 | 0.0 |
| | | | Label Smoothing | **76.4** | 80.8 | 59.0 | 0.0 |
| | | | MaxSup | **76.4** | 79.9 | 62.4 | 0.0 |

0.9 percentage points. These results indicate that MaxSup achieves a better trade-off between many- and medium-shot accuracy. While it does not fully resolve the challenge of imbalanced classification—especially for minority classes—it shows positive effects and offers a promising direction for further extension.

**Corrupted Image Classification** To evaluate the effectiveness of MaxSup on out-of-distribution (OOD) settings, we also conducted experiments on CIFAR10-C benchmark [12] shown in Table 8 following settings in [11]. Table 8 reports the performance of MaxSup and Label Smoothing (LS) on this benchmark using ResNet-50 as the backbone. Specifically, LS yields a better NLL (1.5730 vs. 1.8431), implying more confident probabilistic predictions. However, MaxSup achieves a better ECE (0.1479 vs. 0.1741), indicating better calibration of the predicted confidence scores. These results validate that MaxSup remains effective on OOD datasets, achieving performance comparable to LS across all three metrics.

Table 8: Comparison of MaxSup, Label Smoothing (LS), and standard Cross Entropy (CE) on CIFAR-10-C. Lower is better. Values show mean(std) across three setups.

| Metric | MaxSup | LS | CE |
|---|---|---|---|
| Error (Corr) | 0.362(0.055) | 0.359(0.064) | **0.354**(0.015) |
| NLL (Corr) | 1.770(0.103) | **1.476**(0.111) | 1.819(0.158) |
| ECE (Corr) | **0.145**(0.003) | 0.158(0.015) | 0.260(0.015) |

**Ablation on the Weight Schedule.** We also systematically investigate how different $\alpha$ scheduling strategies impact MaxSup's performance. Empirical results indicate that MaxSup consistently maintains high accuracy across a wide range of schedules, further underscoring its robustness against hyperparameter changes. For additional details and discussions, refer to Section F.

## 4.3 Evaluation on Semantic Segmentation

We further investigate MaxSup's applicability to downstream tasks by evaluating its performance on semantic segmentation using the widely adopted MMSegmentation framework.[6] Specifically, we adopt the Uper-Net [40] architecture with a DeiT-Small backbone, trained on ADE20K. Models pretrained on ImageNet-1K with either MaxSup or Label Smoothing are then fine-tuned under the same cross-entropy objective (Section 4.2.2).

Table 9: Semantic segmentation (multi-scale) on ADE20K using UperNet. All models are pretrained on ImageNet-1K; mIoU reported as percentage.

| Backbone | Method | mIoU |
|---|---|---|
| DeiT-Small | Baseline | 42.1 |
| | Label Smoothing | 42.4 (+0.3) |
| | MaxSup | **42.8** (+0.7) |

Table 9 shows that initializing with MaxSup-pretrained weights yields an mIoU of $42.8\%$, surpassing the $42.4\%$ achieved by Label Smoothing. This improvement indicates that MaxSup fosters more

---

[6]https://github.com/open-mmlab/mmsegmentation

discriminative feature representations conducive to dense prediction tasks. By more effectively capturing class boundaries and within-class variability, MaxSup promotes stronger segmentation results, underscoring its potential to deliver features that are both transferable and highly robust.

## 4.4 Visualization via Class Activation Maps

To better understand how MaxSup fundamentally differs from Label Smoothing (LS) in guiding model decisions, we employ Gradient-weighted Class Activation Mapping (Grad-CAM) [29], which highlights regions most influential for each prediction.

We evaluate DeiT-Small under three training setups: MaxSup (second row), LS (third row), and a baseline with standard cross-entropy (fourth row). As illustrated in Figure 2, MaxSup-trained models more effectively suppress background distractions than LS, which often fixates on unrelated objects—such as poles in "Bird," tubes in "Goldfish," and caps in "House Finch." This behavior reflects LS's error-enhancement mechanism, which can misdirect attention.

Moreover, MaxSup retains a wider spectrum of salient features, as exemplified in the Shark" and Monkey" images, where LS-trained models often omit crucial semantic details (e.g., fins, tails, or facial contours). These findings align with our analysis in Section I, clearly demonstrating that MaxSup preserves richer intra-class information. Consequently, MaxSup-trained models produce more accurate and consistent predictions by effectively leveraging fine-grained object cues. Further quantitative Grad-CAM overlay metrics (e.g., precision and recall for target regions) confirm that MaxSup yields more focused and comprehensive activation maps, further underscoring its overall efficacy.

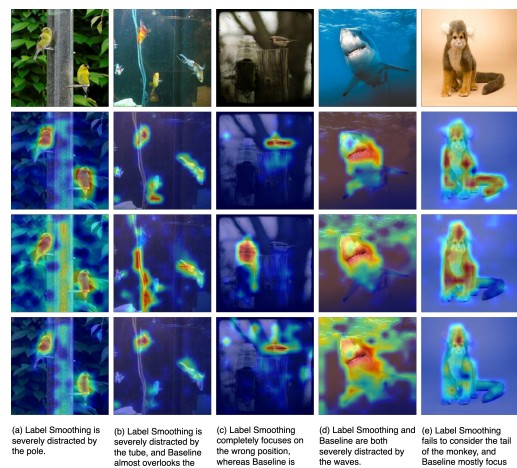

(a) Label Smoothing is severely distracted by the pole.

(b) Label Smoothing is severely distracted by the tube, and Baseline almost overlooks the gold fish at bottom.

(c) Label Smoothing completely focuses on the wrong position, whereas Baseline is distracted by the surrounding objects.

(d) Label Smoothing and Baseline are both severely distracted by the waves.

(e) Label Smoothing fails to consider the tail of the monkey, and Baseline mostly focus on the head forehead.

Figure 2: Grad-CAM [29] visualizations for DeiT-Small models under three training setups: MaxSup (2nd row), Label Smoothing (3rd row), and a baseline (4th row). The first row shows the original images. Compared to Label Smoothing, MaxSup more effectively filters out non-target regions and highlights essential features of the target class, reducing instances where the model partially or entirely focuses on irrelevant areas.

## 5 Conclusion

We examined the shortcomings of Label Smoothing (LS) and introduced Max Suppression Regularization (MaxSup) as a targeted and practical remedy. Our analysis shows that LS can unintentionally heighten overconfidence in misclassified samples by failing to sufficiently penalize incorrect top-1 logits. In contrast, MaxSup uniformly penalizes the highest logit, regardless of prediction correctness, thereby effectively eliminating LS's error amplification. Extensive experiments demonstrate that MaxSup not only improves accuracy but also preserves richer intra-class variation and enforces sharper inter-class boundaries, leading to more nuanced and transferable feature representations and superior transfer performance. Moreover, class activation maps confirm that MaxSup better attends to salient object regions, reducing focus on irrelevant background elements.

**Limitations.** Prior work [23] notes that LS-trained teachers may degrade knowledge distillation [13, 14], and Guo et al. [8] suggests LS accelerates convergence via improved conditioning. Examining MaxSup's potential role in distillation and its overall impact on training dynamics would clarify these underlying effects. Recent studies [33, 7] also show that $\ell_2$ regularization biases final-layer features toward low-rank solutions, raising interesting questions about whether MaxSup behaves similarly.

**Impact.** In practical applications, MaxSup shows strong promise for systems demanding robust generalization and efficient transfer, and we have not observed any additional adverse effects or trade-offs. By offering researchers and practitioners both a clearer understanding of LS's limitations and a straightforward, computationally light, and easily integrable method to overcome them, MaxSup may help guide the development of more reliable and interpretable deep learning models.

## Acknowledgments

This work was supported by the University of Washington Faculty Startup Fund, the Carwein–Andrews Fellowship, the UW GSFEI Top Scholar Award, the U.S. DOT PacTrans sub-center seed funding program, the DFG Research Unit 5336 - Learning to Sense (L2S), and the ELSA – European Lighthouse on Secure and Safe AI funded by the European Union under grant agreement No. 101070617. Views and opinions expressed are however those of the authors only and do not necessarily reflect those of the European Union or European Commission. Neither the European Union nor the European Commission can be held responsible for them.

We thank the anonymous reviewers for their helpful comments.

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

# A Technical Appendices and Supplementary Material

Technical appendices with additional results, figures, graphs and proofs may be submitted with the paper submission before the full submission deadline (see above), or as a separate PDF in the ZIP file below before the supplementary material deadline. There is no page limit for the technical appendices.

# A  Proof of Lemma 3.2

*Proof.* We aim to demonstrate the validity of Lemma 3.2, which states:

$$H(\mathbf{s}, \mathbf{q}) = H(\mathbf{y}, \mathbf{q}) + L_{LS} \tag{10}$$

where $L_{LS} = \alpha \left( H \left( \frac{1}{K}, \mathbf{q} \right) - H(\mathbf{y}, \mathbf{q}) \right)$

Let us proceed with the proof:

We begin by expressing the cross-entropy $H(\mathbf{s}, \mathbf{q})$:

$$H(\mathbf{s}, \mathbf{q}) = -\sum_{k=1}^{K} s_k \log q_k \tag{11}$$

In the context of label smoothing, $s_k$ is defined as:

$$s_k = (1 - \alpha) y_k + \frac{\alpha}{K} \tag{12}$$

where $\alpha$ is the smoothing parameter, $y_k$ is the original label, and $K$ is the number of classes.

Substituting this expression for $s_k$ into the cross-entropy formula:

$$H(\mathbf{s}, \mathbf{q}) = -\sum_{k=1}^{K} \left( (1 - \alpha) y_k + \frac{\alpha}{K} \right) \log q_k \tag{13}$$

Expanding the sum:

$$H(\mathbf{s}, \mathbf{q}) = -(1 - \alpha) \sum_{k=1}^{K} y_k \log q_k - \frac{\alpha}{K} \sum_{k=1}^{K} \log q_k \tag{14}$$

We recognize that the first term is equivalent to $(1 - \alpha) H(\mathbf{y}, \mathbf{q})$, and the second term to $\alpha H(\frac{1}{K}, \mathbf{q})$. Thus:

$$H(\mathbf{s}, \mathbf{q}) = (1 - \alpha) H(\mathbf{y}, \mathbf{q}) + \alpha H \left( \frac{1}{K}, \mathbf{q} \right) \tag{15}$$

Rearranging the terms:

$$H(\mathbf{s}, \mathbf{q}) = H(\mathbf{y}, \mathbf{q}) + \alpha \left( H \left( \frac{1}{K}, \mathbf{q} \right) - H(\mathbf{y}, \mathbf{q}) \right) \tag{16}$$

We can now identify $H(\mathbf{y}, \mathbf{q})$ as the original cross-entropy loss and $L_{LS} = \alpha \left( H \left( \frac{1}{K}, \mathbf{q} \right) - H(\mathbf{y}, \mathbf{q}) \right)$ as the Label Smoothing loss.

Therefore, we have demonstrated that:

$$H(\mathbf{s}, \mathbf{q}) = H(\mathbf{y}, \mathbf{q}) + L_{LS} \tag{17}$$

with $L_{LS}$ as defined in the lemma. It is noteworthy that the original cross-entropy loss $H(\mathbf{y}, \mathbf{q})$ remains unweighted by $\alpha$ in this decomposition, which is consistent with the statement in Lemma 3.2

# B Proof of Theorem 3.3

*Proof.* We aim to prove the equation:

$$L_{LS} = \alpha(z_{gt} - \frac{1}{K}\sum_{k=1}^{K} z_k) \tag{18}$$

Let **s** be the smoothed label vector and **q** be the predicted probability vector. We start with the cross-entropy between **s** and **q**:

$$H(\mathbf{s}, \mathbf{q}) = -\sum_{k=1}^{K} s_k \log q_k \tag{19}$$

With label smoothing, $s_k = (1 - \alpha)y_k + \frac{\alpha}{K}$, where **y** is the one-hot ground truth vector and $\alpha$ is the smoothing parameter. Substituting this:

$$H(\mathbf{s}, \mathbf{q}) = -\sum_{k=1}^{K}[(1 - \alpha)y_k + \frac{\alpha}{K}] \log q_k \tag{20}$$

Expanding:

$$H(\mathbf{s}, \mathbf{q}) = -(1 - \alpha)\sum_{k=1}^{K} y_k \log q_k - \frac{\alpha}{K}\sum_{k=1}^{K} \log q_k \tag{21}$$

Since **y** is a one-hot vector, $\sum_{k=1}^{K} y_k \log q_k = \log q_{gt}$, where $gt$ is the index of the ground truth class:

$$H(\mathbf{s}, \mathbf{q}) = -(1 - \alpha)\log q_{gt} - \frac{\alpha}{K}\sum_{k=1}^{K} \log q_k \tag{22}$$

Using the softmax function, $q_k = \frac{e^{z_k}}{\sum_{j=1}^{K} e^{z_j}}$, we can express $\log q_k$ in terms of logits:

$$\log q_k = z_k - \log(\sum_{j=1}^{K} e^{z_j}) \tag{23}$$

Substituting this into our expression:

$$
\begin{aligned}
H(\mathbf{s}, \mathbf{q}) = & -(1 - \alpha)[z_{gt} - \log(\sum_{j=1}^{K} e^{z_j})] \\
& - \frac{\alpha}{K}\sum_{k=1}^{K}[z_k - \log(\sum_{j=1}^{K} e^{z_j})] \\
= & -(1 - \alpha)z_{gt} + (1 - \alpha)\log(\sum_{j=1}^{K} e^{z_j}) \\
& - \frac{\alpha}{K}\sum_{k=1}^{K} z_k + \alpha \log(\sum_{j=1}^{K} e^{z_j}) \\
= & -(1 - \alpha)z_{gt} - \frac{\alpha}{K}\sum_{k=1}^{K} z_k + \log(\sum_{j=1}^{K} e^{z_j})
\end{aligned} \tag{24}
$$

Rearranging:

$$H(\mathbf{s}, \mathbf{q}) = -z_{gt} + \log(\sum_{j=1}^{K} e^{z_j}) + \alpha[z_{gt} - \frac{1}{K}\sum_{k=1}^{K} z_k] \tag{25}$$

We can identify:

- $H(\mathbf{y}, \mathbf{q}) = -z_{gt} + \log(\sum_{j=1}^{K} e^{z_j})$ (cross-entropy for one-hot vector $\mathbf{y}$)

- $L_{LS} = \alpha[z_{gt} - \frac{1}{K}\sum_{k=1}^{K} z_k]$

Thus, we have proven:

$$H(\mathbf{s}, \mathbf{q}) = H(\mathbf{y}, \mathbf{q}) + L_{LS} \tag{26}$$

Due to the broad usage of CutMix and Mixup in the training recipe of modern Neural Networks, we additionally take their impact into account together with Label Smoothing. Now we additionally prove the case **with Cutmix and Mixup**:

$$L'_{LS} = \alpha((\lambda z_{gt1} + (1 - \lambda)z_{gt2}) - \frac{1}{K}\sum_{k=1}^{K} z_k) \tag{27}$$

With Cutmix and Mixup, the smoothed label becomes:

$$s_k = (1 - \alpha)(\lambda y_{k1} + (1 - \lambda)y_{k2}) + \frac{\alpha}{K} \tag{28}$$

where $y_{k1}$ and $y_{k2}$ are one-hot vectors for the two ground truth classes from mixing, and $\lambda$ is the mixing ratio.

Starting with the cross-entropy:

$$H(\mathbf{s}, \mathbf{q}) = -\sum_{k=1}^{K} s_k \log q_k \tag{29}$$

$$= -\sum_{k=1}^{K}[(1 - \alpha)(\lambda y_{k1} + (1 - \lambda)y_{k2}) + \frac{\alpha}{K}] \log q_k \tag{30}$$

$$= -(1 - \alpha)\sum_{k=1}^{K}(\lambda y_{k1} + (1 - \lambda)y_{k2}) \log q_k - \frac{\alpha}{K}\sum_{k=1}^{K} \log q_k \tag{31}$$

Since $y_{k1}$ and $y_{k2}$ are one-hot vectors:

$$H(\mathbf{s}, \mathbf{q}) = -(1 - \alpha)(\lambda \log q_{gt1} + (1 - \lambda) \log q_{gt2}) - \frac{\alpha}{K}\sum_{k=1}^{K} \log q_k \tag{32}$$

where $gt1$ and $gt2$ are the indices of the two ground truth classes.

Using $q_k = \frac{e^{z_k}}{\sum_{j=1}^{K} e^{z_j}}$, we express in terms of logits:

$$H(\mathbf{s}, \mathbf{q}) = -(1 - \alpha)[\lambda(z_{gt1} - \log(\sum_{j=1}^{K} e^{z_j})) + (1 - \lambda)(z_{gt2} - \log(\sum_{j=1}^{K} e^{z_j}))] \tag{33}$$

$$- \frac{\alpha}{K}\sum_{k=1}^{K}[z_k - \log(\sum_{j=1}^{K} e^{z_j})] \tag{34}$$

Simplifying:

$$H(\mathbf{s}, \mathbf{q}) = -(1-\alpha)[\lambda z_{gt1} + (1-\lambda)z_{gt2}] + (1-\alpha)\log(\sum_{j=1}^{K} e^{z_j}) \tag{35}$$

$$-\frac{\alpha}{K}\sum_{k=1}^{K} z_k + \alpha \log(\sum_{j=1}^{K} e^{z_j}) \tag{36}$$

$$= -(1-\alpha)[\lambda z_{gt1} + (1-\lambda)z_{gt2}] - \frac{\alpha}{K}\sum_{k=1}^{K} z_k + \log(\sum_{j=1}^{K} e^{z_j}) \tag{37}$$

Rearranging:

$$H(\mathbf{s}, \mathbf{q}) = -[\lambda z_{gt1} + (1-\lambda)z_{gt2}] + \log(\sum_{j=1}^{K} e^{z_j}) \tag{38}$$

$$+ \alpha[\lambda z_{gt1} + (1-\lambda)z_{gt2} - \frac{1}{K}\sum_{k=1}^{K} z_k] \tag{39}$$

We can identify:

- $H(\mathbf{y}', \mathbf{q}) = -[\lambda z_{gt1} + (1-\lambda)z_{gt2}] + \log(\sum_{j=1}^{K} e^{z_j})$ (cross-entropy for mixed label $\mathbf{y}'$)

- $L'_{LS} = \alpha[\lambda z_{gt1} + (1-\lambda)z_{gt2} - \frac{1}{K}\sum_{k=1}^{K} z_k]$

Thus, we have proven:

$$H(\mathbf{s}, \mathbf{q}) = H(\mathbf{y}', \mathbf{q}) + L'_{LS} \tag{40}$$

This completes the proof for both cases of Theorem 3.3.

## C   Gradient Analysis

### C.1   New Objective Function

The Cross Entropy with Max Suppression is defined as:

$$L_{\text{MaxSup},t}(x, y) = H\left(y_k + \frac{\alpha}{K} - \alpha \cdot \mathbf{1}_{k=\text{argmax}(\boldsymbol{q})}, \boldsymbol{q}_t^S(x)\right)$$

where $H(\cdot, \cdot)$ denotes the cross-entropy function.

### C.2   Gradient Analysis

The gradient of the loss with respect to the logit $z_i$ for each class $i$ is derived as:

$$\partial_i^{\text{MaxSup},t} = y_{t,i} - y_i - \frac{\alpha}{K} + \alpha \cdot \mathbf{1}_{i=\text{argmax}(\boldsymbol{q})}$$

We analyze this gradient under two scenarios:

**Scenario 1: Model makes correct prediction**

In this case, Max Suppression is equivalent to Label Smoothing. When the model correctly predicts the target class ($\text{argmax}(\boldsymbol{q}) = \text{GT}$), the gradients are:

- For the target class (GT): $\partial_{\text{GT}}^{\text{MaxSup},t} = q_{t,\text{GT}} - \left(1 - \alpha\left(1 - \frac{1}{K}\right)\right)$

- For non-target classes: $\partial_i^{\text{MaxSup},t} = q_{t,i} - \frac{\alpha}{K}$

## Scenario 2: Model makes wrong prediction

When the model incorrectly predicts the most confident class ($\text{argmax}(\boldsymbol{q}) \neq \text{GT}$), the gradients are:

- For the target class (GT): $\partial_{\text{GT}}^{\text{MaxSup},t} = q_{t,\text{GT}} - \left(1 + \frac{\alpha}{K}\right)$

- For non-target classes (not most confident): $\partial_i^{\text{MaxSup},t} = q_{t,i} - \frac{\alpha}{K}$

- For the most confident non-target class: $\partial_i^{\text{MaxSup},t} = q_{t,i} + \alpha\left(1 - \frac{1}{K}\right)$

The Max Suppression regularization technique implements a sophisticated gradient redistribution strategy, particularly effective when the model misclassifies samples. When the model's prediction ($\text{argmax}(\boldsymbol{q})$) differs from the ground truth (GT), the gradient for the incorrectly predicted class is increased by $\alpha(1 - \frac{1}{K})$, resulting in $\partial_{\text{argmax}(\boldsymbol{q})}^{\text{MaxSup},t} = q_{t,\text{argmax}(\boldsymbol{q})} + \alpha(1 - \frac{1}{K})$. Simultaneously, the gradient for the true class is decreased by $\frac{\alpha}{K}$, giving $\partial_{\text{GT}}^{\text{MaxSup},t} = q_{t,\text{GT}} - (1 + \frac{\alpha}{K})$, while for all other classes, the gradient is slightly reduced by $\frac{\alpha}{K}$: $\partial_i^{\text{MaxSup},t} = q_{t,i} - \frac{\alpha}{K}$. This redistribution adds a substantial positive gradient to the misclassified class while slightly reducing the gradients for other classes. The magnitude of this adjustment, controlled by the hyperparameter $\alpha$, effectively penalizes overconfident errors and encourages the model to focus on challenging examples. By amplifying the learning signal for misclassifications, Max Suppression regularization promotes more robust learning from difficult or ambiguous samples.

---

**Algorithm 1** Gradient Descent with Max Suppression (MaxSup)

---

**Require:** Training set $D = \{(\mathbf{x}^{(i)}, \mathbf{y}^{(i)})\}_{i=1}^N$; learning rate $\eta$; number of iterations $T$; smoothing parameter $\alpha$; a neural network $f_\theta(\cdot)$; batch size $B$; total classes $K$.

1: Initialize network weights $\theta$ (e.g., randomly).
2: **for** $t = 1$ to $T$ **do**
    *// Each iteration processes mini-batches of size $B$.*
3:    **for** each mini-batch $\{(\mathbf{x}^{(j)}, \mathbf{y}^{(j)})\}_{j=1}^B$ in $D$ **do**
4:        Compute logits: $\mathbf{z}^{(j)} \leftarrow f_\theta(\mathbf{x}^{(j)})$ for each sample in the batch
5:        Compute predicted probabilities: $\mathbf{q}^{(j)} \leftarrow \text{softmax}(\mathbf{z}^{(j)})$
6:        Compute cross-entropy loss:

$$L_{\text{CE}} \leftarrow \frac{1}{B} \sum_{j=1}^B H\big(\mathbf{y}^{(j)}, \mathbf{q}^{(j)}\big)$$

7:        *// MaxSup component: penalize the top-1 logit*
8:        For each sample $j$:

$$z_{max}^{(j)} = \max_{k \in \{1,\ldots,K\}} z_k^{(j)}, \quad \bar{z}^{(j)} = \frac{1}{K} \sum_{k=1}^K z_k^{(j)}$$

$$L_{\text{MaxSup}} \leftarrow \frac{1}{B} \sum_{j=1}^B \big[z_{max}^{(j)} - \bar{z}^{(j)}\big]$$

9:        Total loss:

$$L \leftarrow L_{\text{CE}} + \alpha\, L_{\text{MaxSup}}$$

10:        Update parameters:

$$\theta \leftarrow \theta - \eta\, \nabla_\theta L$$

11:    **end for**
12: **end for**

---

# D  Pseudo Code

Algorithm 1 presents pseudo code illustrating gradient descent with Max Suppression (MaxSup). The main difference from standard Label Smoothing lies in penalizing the highest logit rather than the ground-truth logit.

# E  Robustness Under Different Training Recipes

We assess MaxSup's robustness by testing it under a modified training recipe that reduces total training time and alters the learning rate schedule. This setup models scenarios where extensive training is impractical due to limited resources.

Concretely, we adopt the **TorchVision V1 Weight** strategy, reducing the total number of epochs to 90 and replacing the cosine annealing schedule with a step learning-rate scheduler (step size = 30). We also set the initial learning rate to 0.1 and use a batch size of 512. This streamlined recipe aims to reach reasonable accuracy within a shorter duration.

As reported in Table 10, MaxSup continues to deliver strong performance across multiple convolutional architectures, generally surpassing Label Smoothing and its variants. Although all methods see a performance decline in this constrained regime, MaxSup remains among the top performers, reinforcing its effectiveness across diverse training conditions.

Table 10: Performance comparison on ImageNet for various convolutional neural network architectures. Results are presented as "mean ± std" (percentage). **Bold** and underlined entries indicate best and second-best, respectively. (*: implementation details adapted from the official repositories.)

| Method | ResNet-18 | ResNet-50 | ResNet-101 | MobileNetV2 |
|---|---|---|---|---|
| Baseline | 69.11±0.12 | 76.44±0.10 | 76.00±0.18 | 71.42±0.12 |
| Label Smoothing | 69.38±0.19 | 76.65±0.11 | 77.01±0.15 | 71.40±0.09 |
| Zipf-LS* | 69.43±0.13 | 76.89±0.17 | 76.91±0.14 | 71.24±0.16 |
| OLS* | 69.45±0.15 | 76.81±0.21 | 77.12±0.17 | 71.29±0.11 |
| **MaxSup** | **69.59**±0.13 | **77.08**±0.07 | **77.33**±0.12 | **71.59**±0.17 |
| Logit Penalty | 66.97±0.11 | 74.21±0.16 | 75.17±0.12 | 70.39±0.14 |

# F  Increasing Smoothing Weight Schedule

Building on the intuition that a model's confidence naturally grows as training progresses, we propose a linearly increasing schedule for the smoothing parameter $\alpha$. Concretely, $\alpha$ is gradually raised from an initial value (e.g., 0.1) to a higher value (e.g., 0.2) by the end of training. This schedule aims to counteract the model's increasing overconfidence, ensuring that regularization remains appropriately scaled throughout.

**Experimental Evidence**    As shown in Table 11, both Label Smoothing and MaxSup benefit from this $\alpha$ scheduler. For Label Smoothing, accuracy improves from 75.91% to 76.16%, while MaxSup sees a more pronounced gain, from 76.12% to 76.58%. This greater improvement for MaxSup ($+0.46\%$) compared to Label Smoothing ($+0.25\%$) corroborates our claim that MaxSup successfully addresses the inconsistent regularization and error-enhancement issues of Label Smoothing during misclassifications.

Table 11: Effect of an $\alpha$ scheduler on model performance. Here, $t$ and $T$ denote the current and total epochs, respectively. The baseline model does not involve any label smoothing parameter ($\alpha$).

| Configuration | Formulation | $\alpha = 0.1$ | $\alpha = 0.1 + 0.1\,\frac{t}{T}$ | Remarks |
|---|---|---|---|---|
| Baseline | – | 74.21 | 74.21 | $\alpha$ not used |
| LS | $\alpha\left(z_{gt} - \frac{1}{K}\sum_k z_k\right)$ | 75.91 | 76.16 | |
| MaxSup | $\alpha\left(z_{max} - \frac{1}{K}\sum_k z_k\right)$ | 76.12 | **76.58** | |

# G  Extended Evaluation of Linear Transferability on Different Datasets

To further demonstrate the substantial improvement in feature representation compared to other methods, we further compare the linear transfer accuracies of different methods on a broader range of datasets in Table 12

Table 12: Validation performance of different methods, evaluated using multinomial logistic regression with l2 regularization. Although Label Smoothing and OLS improve ImageNet accuracy, they substantially degrade transfer accuracy compared to MaxSup. Following [15], we selected from 45 logarithmically spaced values between $10^{-6}$ and $10^5$. Notably, the search range is larger than the search range used in Table 3, thus leading to higher overall accuracies on CIFAR10.

| Datasets | CIFAR10 | CIFAR100 | CUB | Flowers | Foods | Pets |
|----------|---------|----------|-----|---------|-------|------|
| CE | 91.74 | 75.35 | 70.21 | 90.96 | 72.44 | 92.30 |
| LS | 90.14 | 71.28 | 64.50 | 84.84 | 67.76 | 91.96 |
| OLS | 90.29 | 73.13 | **67.86** | 87.47 | 69.34 | 92.21 |
| MaxSup | **91.00** | **73.93** | 67.29 | **88.84** | **70.94** | **92.93** |

## H    Extended Comparison to More Label Smoothing Alternatives

We have included Confidence Penalty [25] and Adaptive Label Smoooothing with Self-Knowledge [20] for an extended comparison in Table 13. We follow their recommended hyperparameter settings: weight coefficient is set to 0.1 for confidence penalty, and the checkpoint with the highest validation accuracy checkpoint is treated as teacher for Adaptive Label Smoothing.

To further address the novelty of our work, we additionally compared our method to recent approaches that also identify and aim to fix the issues of Label Smoothing on misclassified samples [6, 38]. Specifically, the method proposed in [38] aims to mitigate overconfidence via logit normalization during training. With its default hyperparameter settings, it achieves an accuracy of 74.32% on ImageNet with a ResNet-50, which is significantly lower than the 76.91% achieved by standard Label Smoothing. This performance aligns with that of Logit Penalty, which similarly minimizes the global $l_2$-norm of logits and can struggle to match baseline LS performance. We also note that these norm-based methods[6, 38, 4] are often highly sensitive to hyperparameter choices, which can limit their practical applicability.

As shown in 13, MaxSup outperforms all these alternatives on ImageNet with ResNet-50. This aligns with our theoretical analysis that selectively penalizing $z_{max}$ yields a more consistent and effective regularization than penalizing all logits (Confidence Penalty), dynamically smoothing the label distribution with self-knowledge (Adaptive LS), or applying global norm-based penalties (Logit Penalty, Logit Normalization).

Table 13: Comparison of classic convolutional neural networks on ImageNet. Results are reported as "mean $\pm$ std" (percentage). **Bold** entries highlight the best performance; underlined entries mark the second best. (Methods with * denote code adaptations from official repositories; see text for details.)

| Method | ImageNet |
|--------|----------|
| | **ResNet-50** |
| Baseline | $76.41 \pm 0.10$ |
| Label Smoothing | $76.91 \pm 0.11$ |
| Zipf-LS* | $76.73 \pm 0.17$ |
| OLS* | $77.23 \pm 0.21$ |
| **MaxSup** | $77.69 \pm 0.07$ |
| Logit Penalty | $76.73 \pm 0.10$ |
| Logit Normalization [3*] | 74.32 |
| Confidence Penalty | $76.58 \pm 0.12$ |
| Adaptive Label Smoothing* | $77.16 \pm 0.15$ |

## I    Visualization of the Learned Feature Space

To illustrate the differences between Max Suppression Regularization and Label Smoothing, we follow the projection technique of Müller et al. [23]. Specifically, we select three semantically related classes and construct an orthonormal basis for the plane intersecting their class templates in feature space. We then project each sample's penultimate-layer activation vector onto this plane. To ensure the visual clarity of the resulting plots, we randomly sample 80 images from the training or validation set for each of the three classes.

**Selection Criteria**   We choose these classes according to two main considerations:

1. **Semantic Similarity.** We pick three classes that are visually and semantically close.

2. **Confusion.** We identify a class that the Label Smoothing (LS)–trained model frequently misclassifies and select two additional classes involved in those misclassifications (Figure 3c, Figure 4c). Conversely, we also examine a scenario where a class under Max Suppression is confused with others, highlighting key differences (Figure 3d, Figure 4d).

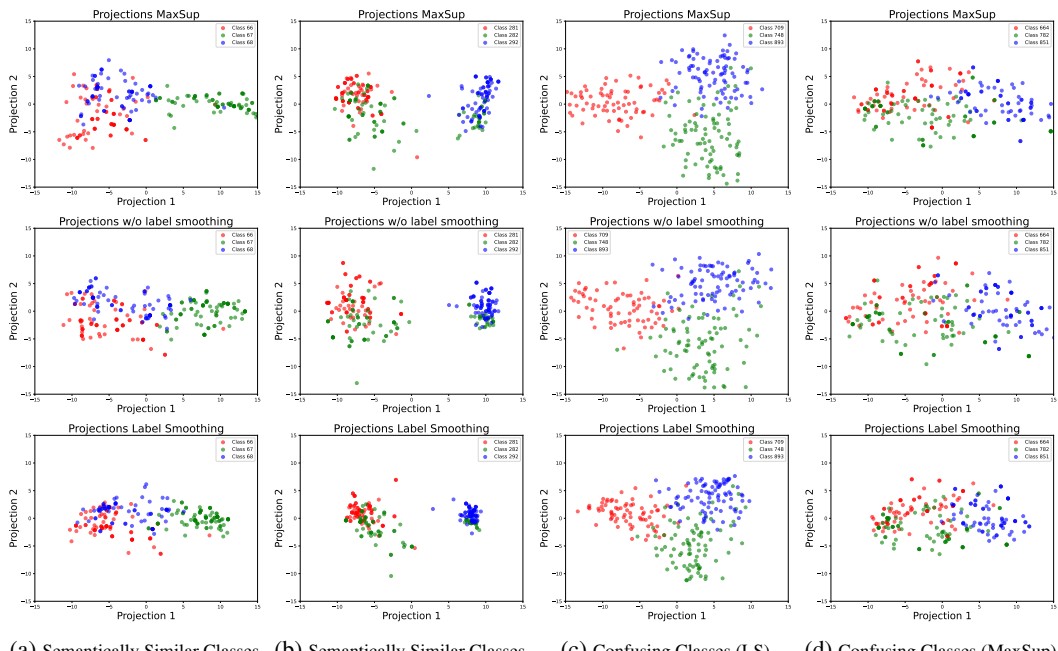

(a) Semantically Similar Classes   (b) Semantically Similar Classes   (c) Confusing Classes (LS)   (d) Confusing Classes (MaxSup)

Figure 3: Visualization of penultimate-layer activations from DeiT-Small (trained with CutMix and Mixup) on the ImageNet **validation** set. The top row shows embeddings for a MaxSup-trained model, and the bottom row shows embeddings for a Label Smoothing (LS)–trained model. In each subfigure, classes are either *semantically similar* or *confusingly labeled*. Compared to LS, MaxSup yields more pronounced inter-class separability and richer intra-class diversity, suggesting stronger representation and classification performance.

**Observations**   As shown in Figures 3 and 4, models trained with **Max Suppression** exhibit:

- **Enhanced inter-class separability.** Distinct classes occupy more clearly separated regions, aligning with improved classification performance.

- **Greater intra-class variation.** Instances within a single class are not overly compressed, indicating a richer representation of subtle differences.

For instance, images of *Schipperke* dogs can differ markedly in viewpoint, lighting, background, or partial occlusions. Max Suppression preserves such intra-class nuances in the feature space, enabling the semantic distances to visually related classes (e.g., Saluki, Grey Fox, or Belgian Sheepdog) to dynamically adjust for each image. Consequently, Max Suppression provides a more flexible, fine-grained representation that facilitates better class discrimination.

## J   Ablation on the Weight Schedule

We conducted an ablation study on the $\alpha$ schedule, as shown in Table 15. The consistently high accuracy across settings demonstrates the robustness of MaxSup. The adaptive $\alpha$ schedule, adopted from [20], further highlights the method's integrity and compatibility with principled design choices.

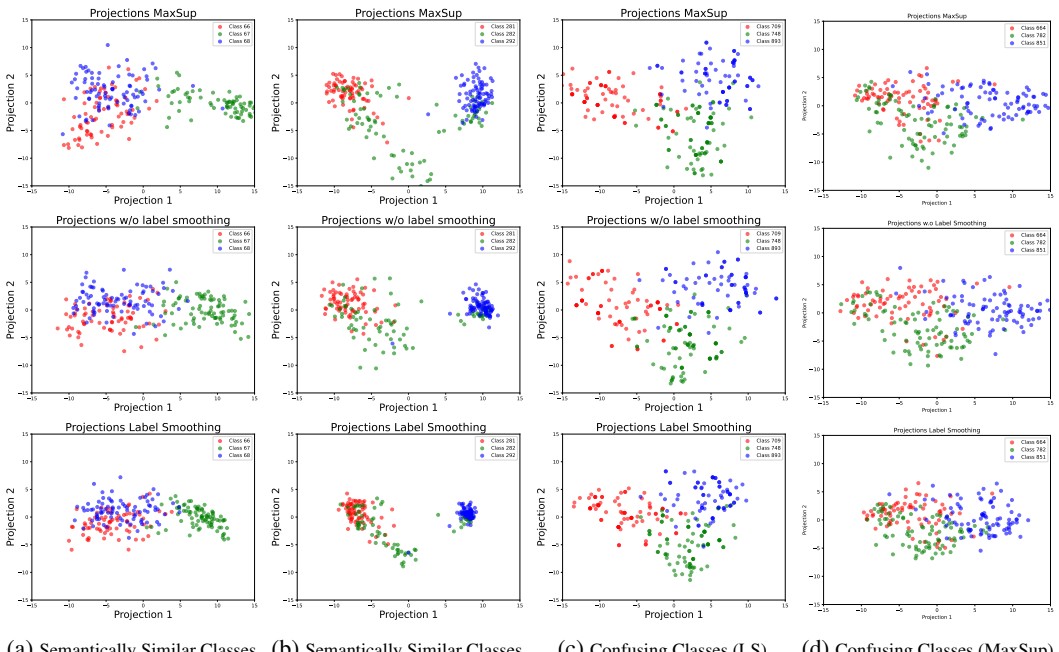

(a) Semantically Similar Classes  (b) Semantically Similar Classes  (c) Confusing Classes (LS)  (d) Confusing Classes (MaxSup)

Figure 4: Visualization of the penultimate-layer activations for DeiT-Small (trained with CutMix and Mixup) on selected ImageNet classes. The top row shows results for a MaxSup-trained model; the bottom row shows Label Smoothing (LS). In (a,b), the model must distinguish *semantically similar* classes (e.g., Saluki vs. Grey Fox; Tow Truck vs. Pickup), while (c,d) involve *confusing categories* (e.g., Jean vs. Shoe Shop, Stinkhorn vs. related objects). Compared to LS, MaxSup yields both improved inter-class separability and richer intra-class variation, indicating more robust representation learning.

Table 14: Feature representation metrics for a ResNet-50 model trained on ImageNet-1K, reported on both Training and Validation sets. We measure intra-class variation ($\bar{d}_{\text{within}}$) and overall average distance ($\bar{d}_{\text{total}}$). Inter-class separability ($R^2$) is calculated as $R^2 = 1 - \frac{\bar{d}_{\text{within}}}{\bar{d}_{\text{total}}}$. Higher values ($\uparrow$) of $\bar{d}_{\text{within}}$ and $R^2$ are preferred.

| Method | $\bar{d}_{\text{within}}$ $\uparrow$ | | $\bar{d}_{\text{total}}$ | | $R^2$ $\uparrow$ | |
|---|---|---|---|---|---|---|
| | **Train** | **Val** | **Train** | **Val** | **Train** | **Val** |
| Baseline | 0.3114 | 0.3313 | 0.5212 | 0.5949 | 0.4025 | 0.4451 |
| LS | 0.2632 | 0.2543 | 0.4862 | 0.4718 | 0.4690 | 0.4611 |
| OLS | 0.2707 | 0.2820 | 0.6672 | 0.6570 | 0.5943 | 0.5708 |
| Zipf's | 0.2611 | 0.2932 | 0.5813 | 0.5628 | 0.5522 | 0.4790 |
| **MaxSup** | **0.2926 (+0.03)** | **0.2998 (+0.05)** | 0.6081 (+0.12) | 0.5962 (+0.12) | 0.5188 (+0.05) | 0.4972 (+0.04) |
| Logit Penalty | 0.2840 | 0.3144 | 0.7996 | 0.7909 | 0.6448 | 0.6024 |

# K   Analysis of Computation Efficiency

Beyond the standard cross-entropy operations, MaxSup only requires:

1. A `max` operation to determine the largest logit ($O(K)$ complexity),
2. A `mean` operation over the $K$-dimensional logit vector, and
3. One subtraction between these two scalars.

Table 15: Ablation study on alpha schedules using ResNet50 on ImageNet1K.

| Schedule | $\alpha = 0.1 + 0.1\frac{t}{T}$ | $\alpha = 0 + 0.1\frac{t}{T}$ | $\alpha = 0.2 + 0.1\frac{t}{T}$ | Adaptive Alpha |
|---|---|---|---|---|
| | 77.65 | 77.62 | 77.43 | 77.70 |

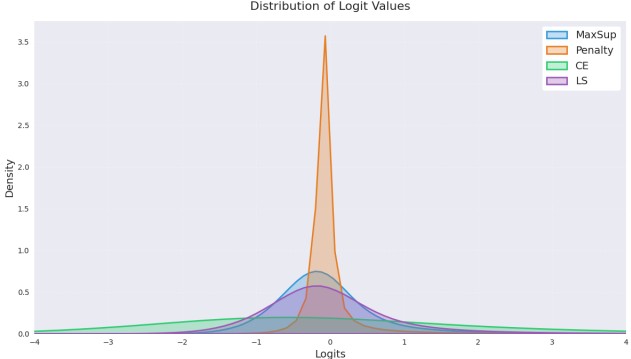

Figure 5: Comparison of logit distributions under different regularizers.

Since $K$ (the number of classes) is usually small (e.g., 1000 for ImageNet-1K), this overhead is minimal compared to the overall forward/backward pass of a deep network. In Table 16, we report the average training time per epoch using a ResNet-50 model on the ImageNet-1K dataset.

Table 16: Average training time per epoch on ImageNet-1K with ResNet-50.

| Method | CE (One-Hot) | CE + LS | CE + MaxSup |
|---|---|---|---|
| **Time/Epoch** | 3 min 51 s | 3 min 52 s | 3 min 51 s |

As seen above, the measured run times are nearly identical across all three configurations. Thus, the additional cost of MaxSup is negligible compared to the total computation for large-scale training.

## L  Logits Visualization

As mentioned in Section 4.1.2, all Label Smoothing variants apply a different penalty on the logits. To illustrate the impact of different methods on the logits, we plot the histogram of logits of ResNet-50 networks trained with each method over the ImageNet validation set, as shown in Figure 5.

