# OpenReview forum: "MaxSup: Overcoming Representation Collapse in Label Smoothing"
_NeurIPS.cc/2025/Conference — NeurIPS 2025 oral_

### Official Review · Reviewer_1zdf · 2025-06-05

**Clarity:** 3
**Significance:** 3
**Originality:** 3
**Rating:** 5
**Confidence:** 4

**Summary:**

The authors propose a simple modification to the widespread label smoothing regulariser: penalise the max logit rather than the ground truth logit, and term it MaxSup. This design choice is based on the observation that label smoothing does not penalise the logit of the prediction when the model is incorrect during training, but rather the logit of the ground truth label.

They demonstrate visually and quantitatively that MaxSup improves the inter-class separability of features, whilst preserving intra-class variation, allowing for improved classification accuracy without sacrificing downstream transferability of features.

Finally, they demonstrate modest but consistent improvements in performance over label smoothing over a wide range of experiments.

**Questions:**

This submission appears to be a revised version of a paper I extensively reviewed previously. The authors had/have addressed earlier concerns effectively, and I am generally satisfied with the current manuscript.

I would like, however, my one concern about discussing related work to be addressed. The authors should also add the underbraces from Eq. 6 to Eq. 2.

**Ethical Concerns:**

["NO or VERY MINOR ethics concerns only"]

**Final Justification:**

I was already mostly satisfied with this submission pre-rebuttal and the authors addressed my only concern about better referencing concurrent work.

**Limitations:**

Yes

**Quality:**

3

**Strengths And Weaknesses:**

## Strengths
- The proposed approach is well motivated by the analysis in Section 3.1. Intuitively, regularising the ground truth logit when the model is wrong is not the best choice, and this is well argued and presented.
- MaxSup itself is a simple, slot-in replacement for label smoothing. This is a big positive for me, as it allows for easier adoption by practitioners, that don't have to worry about implementing a complex system that may end up being fragile to hyperparameter tuning.
- Beyond the simplicity of the approach and its motivation, the wide range of experiments supports the idea that MaxSup will give *consistent/reliable* improvements over label smoothing (compared to more complex approaches that may give more impressive numbers on limited experimental setups, but often exhibit fragility when generalised to a broader range of tasks).
- Experiments on inter-class separability and intra-class variation of features provide additional insight on how MaxSup behaves.
- The visualisations supplement the quantitative results, providing a better intuition for how MaxSup improves over label smoothing.

## Weaknesses
- The authors do not adequately address/discuss concurrent work [1]. Although it is briefly mentioned in L150, the fact that it performs a similar training loss analysis for the (different) task of selective classification would best be discussed in Sec. 2.2 Studies on Label Smoothing.

[1] Xia et al. Towards Understanding Why Label Smoothing Degrades Selective Classification and How to Fix *It ICLR 2025*

---

> ### Author Rebuttal · Authors · 2025-07-31
>
> We are grateful for the reviewer’s thoughtful and constructive feedback. We have carefully considered and incorporated the suggestion into our revision, as detailed below.
>
> > **W1:** The authors do not adequately address/discuss concurrent work [1]. Although it is briefly mentioned in L150, the fact that it performs a similar training loss analysis for the (different) task of selective classification would best be discussed in Sec. 2.2 Studies on Label Smoothing.
>
> We thank the reviewer for pointing this out and are happy to expand our discussion of [1*] in Section 2.2. While [1*] does conduct a related gradient analysis of the training loss, it focuses specifically on the setting of selective classification, and examines a **posthoc** logit normalization technique to mitigate confidence calibration issues. However, this approach addresses only the overconfidence problem of label smoothing (LS), without tackling representation collapse and improving classification accuracy.
>
> In contrast, our work presents a logit-level reformulation of LS that provides a deeper theoretical understanding of why LS amplifies errors and leads to representation collapse. This reformulation motivates MaxSup, a principled and effective alternative that directly regularizes the classifier during training, improves classification accuracy, and resolves the representation collapse issue, which is critical for feature learning and downstream generalization.
>
> We will integrate this expanded comparison into the related work section in the revised version.
>
>
> > **Q1**: The authors should also add the underbraces from Eq. 6 to Eq. 2.
>
> We are glad to incorporate the reviewer's suggestion into our revised version.

---

> > ### Comment · Reviewer_1zdf · 2025-08-03
> >
> > I thank the authors for their response and for incorporating my feedback. I would also like to commend their diligence working on this manuscript over its submission cycle, as it has considerably improved since I first read it.

---

> > > ### Author Response · Authors · 2025-08-05
> > > **Author Follow-up Comment**
> > >
> > > **Dear Reviewer 1zdf,**
> > >
> > > We sincerely appreciate your thoughtful and constructive feedback, and we are especially grateful for your acknowledgment of the improvements made over time.
> > >
> > > Thank you again for your support and engagement throughout the review process.

---

### Official Review · Reviewer_btTx · 2025-06-11

**Clarity:** 3
**Significance:** 3
**Originality:** 3
**Rating:** 4
**Confidence:** 3

**Summary:**

This paper identifies two critical drawbacks of conventional label smoothing (LS): (1) error amplification—incorrect predictions become overconfident, and (2) representation collapse—within-class feature diversity is overly compressed. By decomposing the LS loss at the logit level, the authors show that the standard LS penalty contains both a “regularization term” (which only activates on correct predictions) and an “error-amplification term” (which activates on mispredictions). To remedy this, they propose Max Suppression (MaxSup): instead of penalizing the ground-truth logit, the loss penalizes the current maximum logit. This simple modification ensures that both correct and incorrect predictions are regularized evenly. Extensive experiments on ImageNet classification, downstream segmentation and fine-tuning tasks, and Grad-CAM visualizations demonstrate that MaxSup consistently improves top-1 accuracy (by 0.4–1.0%), preserves within-class diversity, and boosts segmentation mIoU compared to LS and its variants.

**Questions:**

- How does MaxSup scale when the number of classes K is extremely large (e.g., ≥1M)? Have you considered approximate or sparse strategies to find the max logit efficiently?

- If the training set contains mislabeled examples, does MaxSup inadvertently amplify noise? Would a confidence-aware or adaptive suppression strategy help?

- Have you tried integrating MaxSup with teacher–student distillation frameworks? How does it affect the quality of distilled student models?

**Ethical Concerns:**

["NO or VERY MINOR ethics concerns only"]

**Final Justification:**

I appreciate that the authors have provided many detailed experiments to demonstrate the effectiveness of their method across different scenarios. However, the visualization (Figure 1) must be carefully revised. As currently presented, Figure 1 shows that the proposed method performs worse than Label Smoothing, which raises concerns about the validity of MaxSup. Overall, I believe this paper is at the borderline of acceptance—it could go either way. Given the amount of work the authors have put into it, I am inclined toward a borderline accept.

**Limitations:**

- Evaluated only on vision classification and segmentation; its effectiveness in NLP, multimodal tasks, or reinforcement learning remains untested.

- The theoretical analysis is limited to static logit decomposition; there is no formal study of training dynamics or convergence rates under MaxSup.

- Robustness to extreme label noise, long-tail distributions, adversarial attacks, or online learning scenarios is not assessed.

- Don't add too many underlines, as it affects the overall aesthetics of the paper.

**Paper Formatting Concerns:**

- Tables 4 and 5 have misaligned column headers; ensure uniform column widths and clarify any “*” annotations in the captions.

**Quality:**

3

**Strengths And Weaknesses:**

Strengths:

- The logit-level decomposition cleanly separates the beneficial and harmful components of LS, revealing an often-overlooked error amplification effect.

- MaxSup requires only a one-line change to the loss function and plugs directly into existing training pipelines without additional architecture or hyperparameter complexity.

- Results cover multiple backbones (ResNet variants, MobileNetV2, DeiT-S), diverse tasks (ImageNet classification, CIFAR linear probes, ADE20K segmentation, fine-grained recognition), and report means ± standard deviations, supporting strong reproducibility.

- Metrics such as within-class variance, R² transfer accuracy, and Grad-CAM maps all indicate that MaxSup maintains richer intra-class representations and focuses more accurately on relevant image regions.

- The appendix and NeurIPS checklist provide full training details, hyperparameter schedules, and a link to code, facilitating community adoption.


Weaknesses:

- In the introduction section of the paper, it is stated that the left part of Figure 1 shows a feature collapse. However, in reality, MaxSup also fails to learn class 282 properly, and the features learned for this class are completely scattered—performing even worse than the result of label smoothing.

- Experiments do not cover scenarios with high label noise, severe class imbalance, or adversarial examples—settings where LS variants often differ markedly.

- Evaluation is largely confined to LS variants and logit penalties. It would be stronger to compare against related techniques such as self-knowledge distillation, label perturbation, or mixup-based regularizers.

- Although the authors discuss the weight α schedule in the appendix, a systematic sensitivity analysis across a wide α range (e.g., α→0 or α→1 extremes) is missing.

- MaxSup requires identifying the maximum logit for each example. In tasks with extremely large label sets (e.g., million-class classification or retrieval), the computational and memory overhead of that max operation may become nontrivial.

---

> ### Author Rebuttal · Authors · 2025-07-31
>
> We sincerely thank the reviewer for the valuable feedback and deeply appreciate the time and effort invested in reviewing our work. Below, we address the mentioned concerns and suggestions in detail.
>
> > **W1:** Clarification of Figure 1.
>
> - We focus the **severely reduced intra-class variation** caused by label smoothing [13, 26, 37]. This collapse leads to tightly clustered features for each class, which may seem beneficial but is known to **harm downstream tasks** such as transfer learning. **MaxSup explicitly preserves intra-class variation**, as can be observed for classes 281, 282, and 292.
>
> - While class 282 appears more scattered under MaxSup, the **inter-class separation**—particularly between classes 281 and 282—is visibly better (smaller overlapped area) under MaxSup than under ls or ce. This supports the improved classification performance we report and confirms that MaxSup **achieves larger inter-class separation while preserving the intra-class variation**.
>
> > **W2**: Experiments with high label noise, severe class imbalance, or adversarial examples—settings.
>
> We emphasize that $L_{\text{maxsup}}$ **is orthogonal to** strategies for handling these challenges, and can be readily combined with them.
>
> For imbalanced data, we have conducted comprehensive experiments on CIFAR10-LT with varying imbalance ratios (50, and 100) following settings in [1*], as shown in Table R1. Across all imbalance ratios and splits (val/test), MaxSup consistently outperforms both the baseline and LS.
>
> For out-of-distribution (OOD) settings, we conducted experiments on CIFAR10-C using ResNet-50 shown in Table R2 following settings in [2*]. These results validate that **MaxSup remains effective on OOD datasets, achieving performance comparable to LS**.
>
> **Table R1**: Comparison of overall accuracy (%) (jointly considering many-shot, median-shot, and low-shot top-1 accuracy).
>
> | **Dataset**           | **Split** | **Imbalance Ratio** | **Backbone** | **Focal Loss** | **+LS**  | **+MaxSup** |
> |-----------------------|-----------|----------------------|--------------|----------------|--------|------------|
> | Long-tailed CIFAR-10  | val       | 50                   | Resnet32     | 77.4           | 81.2   | **82.1**   |
> | Long-tailed CIFAR-10  | test      | 50                   | Resnet32     | 76.8           | 80.5   | **81.4**   |
> | Long-tailed CIFAR-10  | val       | 100                  | Resnet32     | 75.1           | 76.6   | **77.1**   |
> | Long-tailed CIFAR-10  | test      | 100                  | Resnet32     | 74.7           | 76.4   | **76.4**   |
>
> **Table R2**: Comparison of MaxSup and LS on CIFAR10-C using Resnet50 as backbone. Lower value is better.
>
> | Metric         | MaxSup | LS |
> |----------------|--------|------------------|
> | Error (Corr)   | **0.3951** | **0.3951**           |
> | NLL (Corr)     | 1.8431 | **1.5730**           |
> | ECE (Corr)     | **0.1479** | 0.1741           |
>
> [1*] Tang, K., Huang, J., & Zhang, H. Long-tailed classification by keeping the good and removing the bad momentum causal effect. NeurIPS 2020.
>
> [2*] Heinonen, M., Tran, B. H., Kampffmeyer, M., and Filippone, M. "Robust Classification by Coupling Data Mollification with Label Smoothing." AISTATS 2025.
>
> > **W3**: Comparison to related techniques such as self-knowledge distillation, label perturbation, or mixup-based regularizers.
>
> We have trained ResNet-50 with two representative self-knowledge distillation techniques using ResNet-50 on ImageNet, and below are the results:
>
> | Method                    | Accuracy (%)        |
> |---------------------------|---------------------|
> | Baseline                  | 76.41 ± 0.10        |
> | Label Smoothing           | 76.91 ± 0.11        |
> | MaxSup                    | **77.69 ± 0.07**    |
> | TF-KD$_{reg}$ [3*]         | 76.42 ± 0.09        |
> | DDSGD [4*]                 | 77.35 ± 0.09        |
>
> Notably, **DDSGD slows down training by 2×** since it always reuses half of the previous batch.
>
> [3*] Yuan, Li, et al. "Revisiting knowledge distillation via label smoothing regularization." CVPR 2020.
>
> [4*] Shen, Yiqing, et al. "Self-distillation from the last mini-batch for consistency regularization." CVPR 2022.
>
> > **W4**: Sensitivity analysis across a wide α range (e.g., α→0 or α→1 extremes).
>
> We appreciate the reviewer’s interest in the sensitivity of α. We note that extreme values (e.g., α → 0 or α → 1) are **less informative and rarely used in practice**: As α → 0, the method reduces to standard cross-entropy, which we already include as a baseline; As α → 1, the supervision signal vanishes, leading to  degraded performance.
>
> For ResNet-50 trained with MaxSup on ImageNet, when α = 0.5, the top-1 accuracy is **76.11%**, while at α = 0.9, it drops sharply to **68.38%**. Our analysis thus focuses on the practical range of α where MaxSup meaningfully improves performance.
>
> > **W5**: In tasks with extremely large label sets, the computational and memory overhead of max operation.
>
> We appreciate the reviewer's concern on the scalability of MaxSup. However, even in extreme scenarios, the **computation required to find the maximum logit** is **negligible** compared to the **total cost of a forward pass** in modern deep networks.
>
> Let $K$ denote the number of classes. Identifying the maximum logit requires approximately $1 \times 10^6$ FLOPs for $K = 10^6$. This amounts to only: $0.001 \text{ GFLOPs}$. By comparison, a single forward pass of ResNet-50 on a 224×224 input image requires about **4 GFLOPs**. Therefore, it adds **virtually no measurable runtime overhead**.
>
> > **Q1**: How does MaxSup scale when the number of classes K is extremely large?
>
> This is not an issue for MaxSup, please see our response to **W5**.
>
> > **Q2**: If the training set contains mislabeled examples, does MaxSup inadvertently amplify noise?
>
> Thank you for the thoughtful question. Since MaxSup penalizes overconfidence in the top-1 prediction **regardless of the ground-truth label**, it does not specifically amplify label noise. In fact, by discouraging overly confident predictions, MaxSup can potentially reduce the harmful impact of mislabeled examples.
>
> > **Q3**: MaxSup with teacher–student distillation frameworks.
>
> Following [21], we evaluated the effectiveness of distillation from ResNet-50 to AlexNet on CIFAR-10. Both teacher and student models are trained for 100 epochs using the Adam optimizer with a fixed learning rate of 0.001. The distillation process is conducted with a fixed temperature of 1.0 and an interpolation factor α of 0.1 across all settings:
>
> |                  | Teacher CE | Teacher LS | Teacher MS | Student Baseline | Student from CE | Student from LS | Student from MS |
> |------------------|------------|------------|------------|------------------|------------------|------------------|------------------|
> | **Score**        | 88.76      | 88.96      | 89.47      | 76.66            | 77.95            | 77.41            | 77.84            |
>
> Although the student distilled from the teacher using MaxSup (MS) does not outperform the student distilled with cross-entropy (CE), it surpasses the performance of the student distilled with label smoothing (LS). This result aligns with Table 2, which shows that MaxSup effectively mitigates the variation collapse issue.
>
> > **L1**: Effectiveness in NLP, multimodal tasks, or reinforcement learning remains untested.
>
> We acknowledge the reviewer’s point and agree that a full exploration across domains is valuable future work. While our focus is on computer vision due to the well-documented issues of label smoothing in this domain, we provide preliminary evidence that MaxSup also generalizes to **natural language processing**. Specifically, we train a 12-layer Transformer [1] on the IWSLT 2014 German to English dataset, following the training setup of fairseq repository. The detokenized SacreBLEU scores of 3 runs are compared below. While this enhancement may not appear substantial, it likely stems from the constraints of downstream tasks. Nevertheless, the improvement is statistically significant, as it exceeds the standard deviation.
>
> **Table: Comparison of Label Smoothing and MaxSup on IWSLT 2014 German to English Dataset**
>
> | Model                       | Param. | Method           | BLEU Score     |
> |----------------------------|--------|------------------|----------------|
> | Transformer [1*] | 38M    | Label Smoothing  | 34.4 ± 0.07    |
> |                            |        | MaxSup           | 34.6 ± 0.09    |
>
> [1*] Vaswani, Ashish, et al. "Attention is all you need." Advances in neural information processing systems 30 (2017).
>
> > **L2**: No formal study of training dynamics or convergence rates under MaxSup.
>
> We appreciate the insightful comment. As briefly mentioned in our Limitation section, Guo et al. [9] demonstrate that label smoothing (LS) accelerates convergence through a conditioning number analysis. A similar approach could be applied to analyze the training dynamics of MaxSup. However, this lies beyond the current scope and is left as a promising direction for future work.
>
> > **L3**: Robustness to extreme label noise, long-tail distributions, adversarial attacks, or online learning scenarios is not assessed.
>
> MaxSup applies the intended regularization when the model makes incorrect predictions, i.e., it consistently suppresses the max logit position even when it is not the class label position. This is independent of the assumption of label noise, or long-tail distributions. For the results on imbalanced dataset and dataset with noisy labels, please refer to our response to **w2**.
>
> > **Paper Formatting Concerns**.
>
> We appreciate the reviewer's suggestion and are glad to improve this aspect in our revised version.

---

> ### Comment · Reviewer_btTx · 2025-08-01
>
> Thank you for the detailed response. However, my primary concern (Question 1) has not been fully addressed. A well-structured feature space is expected to exhibit small intra-class variance and large inter-class distance, which is not clearly demonstrated in Figure 1. I suggest that the authors consider replacing the illustration with a more convincing one, or alternatively, provide additional visualizations in the appendix. I will not change my score.

---

> > ### Author Response · Authors · 2025-08-03
> > **Reply to the follow-up comment**
> >
> > We sincerely thank Reviewer btTx for the continued, constructive feedback.
> >
> > We would like to clarify a misunderstanding regarding the intended structure of the feature space. As discussed in the introduction (lines 27–29), prior work [13, 26, 37] has shown that Label Smoothing (LS) often leads to **overly tight feature clusters**, reducing intra-class variation and thereby **harming generalization and transferability** [5]. These works demonstrate that **minimizing intra-class variance is not always beneficial**, which contradicts the assumption that “a well-structured feature space is expected to exhibit small intra-class variance and large inter-class distance.”
> >
> > Instead, these findings support the view that a desirable feature space **maintains sufficient intra-class variation** (which supports transfer learning) while also **ensuring strong inter-class separation** (which supports classification performance).
> >
> > The visualization in Figure 1 is **consistent with the quantitative results** in Table 2: MaxSup retains intra-class variation similar to standard cross-entropy, while achieving greater inter-class separation.
> >
> > To reinforce this point, we have already included **additional visualizations in Figure 3 (Appendix)**. These provide further examples across classes and show that MaxSup consistently **preserves intra-class variation comparable to cross-entropy**, whereas Label Smoothing significantly reduces it. At the same time, MaxSup achieves larger inter-class separation than cross-entropy, which contributes directly to the improved classification performance reported in our results.
> >
> > Finally, we note that **intra-class variation naturally varies across classes and samples**, as can be seen in Figure 3. The more scattered distribution observed for class 282 under MaxSup reflects this variability and does **not** indicate a failure of the method.
> >
> > We hope these clarifications and revisions address your concern. Thank you again for your thoughtful review.

---

### Official Review · Reviewer_wVqR · 2025-06-30

**Clarity:** 3
**Significance:** 2
**Originality:** 2
**Rating:** 4
**Confidence:** 3

**Summary:**

This paper identifies and addresses shortcomings of Label Smoothing (LS), a widely adopted regularization technique. The authors analytically decompose the LS-induced loss, revealing two key components: a beneficial regularization term and a detrimental "error-amplification" term that leads to overconfidence in misclassified samples and feature representation collapse. To overcome these issues, the paper proposes Max Suppression (MaxSup). MaxSup applies uniform regularization by penalizing the top-1 logit rather than the ground-truth logit, eliminating the error amplification effect. Through extensive experiments on large-scale image classification (ImageNet) , semantic segmentation , and downstream tasks, MaxSup is shown to be a more robust alternative to LS, consistently reducing overconfidence, preserving richer feature representations, restoring intra-class variation, and sharpening inter-class boundaries.

**Questions:**

The paper mentions that both MaxSup and standard LS employ a linearly increasing $\alpha$-scheduler for stability. While the robustness to different schedules is noted, a discussion on optimal strategies for dynamically adjusting $\alpha$ (similar to adaptive LS variants) specifically for MaxSup could be valuable. Did the authors observe significantly different results when not using $\alpha$-scheduler. Recently, [2] found that coupling label smoothing and data mollification through schedulers are beneficial. The authors could compare this recent method to really see the effect of the $\alpha$-scheduler and to disentangle its effect from the main proposed method.

Ref:
[2] Heinonen et al. Robust Classification by Coupling Data Mollification with Label Smoothing. AISTATS 2025.

**Ethical Concerns:**

["NO or VERY MINOR ethics concerns only"]

**Final Justification:**

The authors provided some additional experiments that addressed some of my concerns. However, my main concern of the novelty compared to [1*] still remains. Thus, I would like to increase the score by one.

**Limitations:**

Please see the Weaknesses. Overall, the main limitations are two fold: (1) the novelty is limited compared to the recent work in the literature, (2) the analysis is mainly based on intuitions and complicated empirical evidence.

**Quality:**

2

**Strengths And Weaknesses:**

**Strengths**

- The paper is well-written and organized.
- The proposed method, MaxSup, is simple and introduces negligible computational overhead, making it easily integratable into existing training pipelines.
- The paper provides comprehensive experimental results across various architectures (ResNet, MobileNetV2, DeiT-S) and tasks (ImageNet classification, fine-grained classification, semantic segmentation, transfer learning). MaxSup consistently outperforms LS and its variants in terms of top-1 accuracy, preserves intra-class diversity , and improves transferability.

**Weaknesses**

- Indeed, the novelty is quite limited as the issues with LS on misclassified samples have been considered extensively in the prior work [1]. The authors should at least empirically compare this work.
- While the analytical decomposition of LS is detailed, the theoretical implications of penalizing the top-1 logit beyond its direct effect on error amplification could be further explored. For instance, how does it influence the geometry of the learned feature space in a deeper mathematical sense, beyond the empirical observations of preserving intra-class diversity? At the moment, the authors mainly provide some intuitions and empirical evidence. Although the authors provide some analysis in experiments, they are still complicated and consist of many compounded factors to really confirm the intuitions. It would be great if the authors could provide some simple, controlled experiments on even synthetic data.
- The paper isolates MaxSup's contribution by disabling CutMix and Mixup. A discussion or additional experiments on how MaxSup interacts when combined with other common regularization techniques and data augmentations (which are often used in practice) could provide more practical insights.

Ref:
[1] Xia et al. Towards Understanding Why Label Smoothing Degrades Selective Classification and How to Fix It. ICLR 2025

---

> ### Author Rebuttal · Authors · 2025-07-31
>
> We sincerely thank the reviewer for the detailed and constructive feedback. In response to the reviewer’s concerns, we address the mentioned concerns below:
>
> > **W1 – Indeed, the novelty is quite limited as the issues with LS on misclassified samples have been considered extensively in the prior work [1]. The authors should at least empirically compare this work.**
>
> ### 1. Conceptual distinctions
>
> We already **cite and briefly discuss** the distinctions with [1*] ([35] in our paper, lines 149–151), and are happy to expand on this comparison in more detail below.
>
> While [1*] performs a related gradient analysis, it is conducted in the context of **selective classification** and evaluates a **post-hoc** logit normalization technique (previously suggested in [2*]) to address overconfidence **at inference time**. This approach:
>
> - targets **confidence calibration**, not **representation collapse**,
> - does **not improve classification accuracy**, and
> - relies on **additional hyperparameter tuning**.
>
> In contrast, our **logit-level reformulation** of the training objective reveals the root cause of **error amplification** in LS. This decomposition leads directly to **MaxSup**, a principled and training-time regularization method that:
>
> - mitigates overconfidence **during training**,
> - improves **classification accuracy**,
> - and **resolves representation collapse**, a critical issue for transferable feature learning.
>
> ### 2. Empirical comparison (more results)
>
> To address the reviewer's concern, we additionally compared to [3*], which proposed to mitigate overconfidence via **logit normalization during training**. With the default hyperparameter, it merely achieves an accuracy of 74.322 (lag far behind the 76.91 accuracy of label smoothing). The performance aligns with that of Logit Penalty [4] in our Table 4, which similarly minimizes the global L2-norm of logits. We also note that **norm-based methods** [1*, 3*, 4] are **highly sensitive** to hyperparameter choices, which limits their practical applicability. We will integrate this expanded comparison in Appendix H in the revised version.
>
> ### 3. Highlighting our novelty
>
> Our method is not merely incremental; it represents a **substantial rethinking** of Label Smoothing by:
>
> 1. **Theoretically uncovering and explicitly characterizing** the "error amplification" mechanism within the LS-induced loss.
> 2. **Introducing MaxSup**, a simple yet effective modification of penalizing the top-1 logit rather than the ground-truth logit, thus resolving critical limitations of LS during training.
> 3. Demonstrating consistent, robust improvements across various model architectures (CNNs, Vision Transformers), datasets, and tasks (classification, segmentation, fine-grained recognition), highlighting the generalizability and practical value of MaxSup.
>
>
> ### References
>
> [1*] Xia et al. Towards Understanding Why Label Smoothing Degrades Selective Classification and How to Fix It. ICLR 2025
>
> [2*] Luís Felipe Prates Cattelan and Danilo Silva. How to fix a broken confidence estimator: Evaluating post-hoc methods for selective classification with deep neural networks. In UAI, 2024.
>
> [3*] Wei, Hongxin, et al. "Mitigating neural network overconfidence with logit normalization." International conference on machine learning. PMLR, 2022.
>
>
> > **W2**: Regarding the theoretical implications of penalizing the top-1 logit **beyond the impact on error amplification**, such as the influence on the geometry of the learned feature space or some simple, controlled experiments on even synthetic data.
>
> We appreciate the reviewer’s thoughtful comment. While we agree that a deeper theoretical analysis of the geometric effects of penalizing the top-1 logit would be valuable, we respectfully note that this direction **goes beyond the scope of our current study**. Our primary focus is to address the **well-established representation collapse issue** induced by label smoothing [13, 26, 37]. We demonstrate the effectiveness of MaxSup in mitigating this issue through widely adopted qualitative [21] and quantitative [13] evaluations, which we believe are **sufficient for supporting our claims**.
>
> Additionally, we would like to emphasize that representation collapse is a phenomenon that mainly arises in **complex, real-world data settings**. In contrast, prior work such as Neural Collapse [42] has shown that in **simplified or synthetic settings**—where models achieve near-zero training error—representations often collapse naturally, regardless of the loss function used. Therefore, we believe that **synthetic experiments may not meaningfully reflect the challenges** our method is designed to address.
>
>
> > **W3**: The paper isolates MaxSup's contribution by disabling CutMix and Mixup. A discussion or additional experiments on how MaxSup interacts when combined with other common regularization techniques and data augmentations (which are often used in practice) could provide more practical insights.
>
> We appreciate the reviewer’s valuable suggestion. In fact, we have  provided **an extended logit-level analysis that incorporates CutMix and Mixup**, which is documented in the **Appendix (lines 641–657)**. As shown in Equation (39), MaxSup can be naturally extended to these settings by replacing the weighted average of two ground-truth logits with the maximum logit value.
>
> Additionally, we actually performed experiments with CutMix and Mixup that we decided to leave in order to not overload the presentation. Nonetheless, for completeness, we share below the results with CutMix and Mixup enabled on DeiT-Small:
>
> | Model (Touvron et al., 2021) | Method          | Mean  | Std   |
> |-----------------------------|------------------|--------------------------|---------------------------|
> | DeiT-Small                  | Baseline with CutMix&Mixup        | 79.69                    |                            0.11      |
> |                             | Label Smoothing  | 79.81 (+0.12)            |                             0.09  |
> |                             | Zipf-LS          | 79.88 (+0.19)            |                             0.08  |
> |                             | OLS              | 79.95 (+0.27)            |                            0.12  |
> |                             | MaxSup           | 80.16 (+0.47)            |                                 0.09  |
>
> These results demonstrate that MaxSup continues to provide consistent improvements even when used in conjunction with strong data augmentations like CutMix and Mixup.
>
> > **Q1**: 1) Optimal strategies for dynamically adjusting (similar to adaptive LS variants) specifically for MaxSup. 2) Robustness to $\alpha$-scheduler. 3) Comparison to [2] which **couples label smoothing and data mollification** through schedulers.
>
> We appreciate the suggestion to explore optimal strategies for dynamically adjusting MaxSup parameters, such as via adaptive α-schedulers. As shown in Table 14 (Appendix), incorporating the α-scheduler from [18] yields moderate additional improvements. However, our work primarily aims to address the *core limitations* of label smoothing—namely, **error amplification** and **representation collapse**—rather than optimizing performance through scheduler tuning. We view the design of optimal α-scheduling strategies as **orthogonal** and consider it a valuable direction for future work.
>
> Importantly, as demonstrated in Table 8, MaxSup consistently outperforms standard label smoothing **with or without** an α-scheduler. While both methods benefit from scheduling, MaxSup’s gains are **robust** and not reliant on this tuning.
>
> Regarding [2*], we note that while the paper also modifies label smoothing, it tackles a *different problem* by **coupling LS with data mollification**. This coupling leads to a different formulation and focus compared to our method. While a detailed comparison **falls outside the scope of our paper**, we will cite [2*] in the revised version to acknowledge its relevance in the broader context.

---

> > ### Author Response · Authors · 2025-08-05
> > **Author Follow-Up Comment**
> >
> > Dear Reviewer wVqR,
> >
> > We are writing to kindly follow up on our rebuttal to your review. We have carefully considered and thoroughly addressed each of your concerns, particularly regarding the distinctions from prior work [1*], the scope and focus of our study, and the additional experimental validations provided.
> >
> > We have not yet received further feedback, and we would be happy to respond to any further questions or clarifications. Otherwise, we hope our previous response has adequately addressed your concerns, clearly articulated the novelty of our approach relative to [1*], and provided sufficient experimental evidence demonstrating the effectiveness of MaxSup in resolving the key issues of label smoothing discussed in this work.
> >
> > Thank you again for your time and thoughtful contributions to the review process.

---

> > > ### Comment · Reviewer_wVqR · 2025-08-05
> > > **Official Comment by Reviewer wVqR**
> > >
> > > Thank you to the authors for the rebuttal. I've read it and the other reviews carefully.
> > >
> > > The new experiments and insights are useful, so I will be increasing my score.
> > >
> > > Best regards,
> > > Reviewer wVqR

---

### Official Review · Reviewer_cjw9 · 2025-07-03

**Clarity:** 4
**Significance:** 3
**Originality:** 3
**Rating:** 5
**Confidence:** 3

**Summary:**

This paper analyzed the shortcomings of Label Smoothing (newly denoted as error amplification). Building on the foundation, the authors proposed Max Suppression (MaxSup), a modified label smoothing that penalizes only the top-1 logit rather than the ground-truth logit

**Questions:**

**Q1.** The proposed method operates effectively when the CE with Hard Labels decreases sufficiently to recover $z\_{gt}$ from $z\_{max}$ for incorrect samples. How about an imbalanced dataset or out-of-distribution? In those cases, $\mathcal{L}\_{MaxSup}$ would not be effective to minor classes

**Q2.** How about integrating the proposed method into other label smoothing methods? It would be synergized enough to increase the model performance

---
**Things to improve the paper that did not impact the score**

* [line 237, 239] wrong subscript ($\bar{d}$within)

* [line 269, 272] missing white space (momentum0.9, epochs60)

**Ethical Concerns:**

["NO or VERY MINOR ethics concerns only"]

**Final Justification:**

As I referred in the official comment, authors have demonstrated the versatility of the proposed method in various environments, and this result will be helpful to future work in LT and OOD fields. As the authors addressed the concerns about the possibility of extension to various environments, I will maintain my rating of Accept

**Limitations:**

yes

**Quality:**

4

**Strengths And Weaknesses:**

**Strengths**

**S1.** This paper figured out the problem of the commonly used method in this field, and developed a simple yet effective method to solve the discovered problem

**S2.** To overcome the simplicity of the proposed method, the authors thoroughly validated the proposed method in various perspectives, such as theorems and proofs, ablation study, gradient analysis, and so on

**S3.** The experimental environment was set up to verify the superiority and versatility of the proposed method sufficiently, and the experimental results were also well analyzed

**Weaknesses**

There is no weakness in the paper that would justify its rejection

---

> ### Author Rebuttal · Authors · 2025-07-31
>
> We sincerely thank the reviewer for acknowledging the significance of our contribution and valuable suggestions. We address the mentioned questions in detail, as outlined below:
>
> > **Q1**: The proposed method operates effectively when the CE with Hard Labels decreases sufficiently to recover $z_{gt}$ from $z_{max}$ for incorrect samples. How about an imbalanced dataset or out-of-distribution? In those cases, $L_\text{maxsup}$ would not be effective to minor classes?
>
> Thank you for the insightful question. In our study, we had not explicitly addressed class imbalance or out-of-distribution (OOD). However, we emphasize that **$L_{\text{maxsup}}$ is orthogonal to existing strategies for handling these challenges**, and can be readily combined with standard technique. For instance, reweighting the total loss by the inverse class frequency can mitigate bias toward majority classes, while entropy-based regularization may enhance robustness under OOD conditions where predictions are often overconfident.
>
> To specifically address the concern on imbalanced data, we have conducted comprehensive experiments on CIFAR10-LT with varying imbalance ratios (50, and 100) following settings in [1*], as shown in Table R1. The evaluation compares three setups: Focal Loss, Focal Loss + Label Smoothing (LS), and Focal Loss + MaxSup. Across all imbalance ratios and splits (val/test), MaxSup consistently outperforms both the baseline and LS in overall accuracy, which jointly reflects the many-shot, medium-shot, and low-shot (minor class) performance. For example, at an imbalance ratio of 50 on the test split, MaxSup achieves 81.4% accuracy, outperforming Focal Loss (76.8%) by 4.6 percentage points, and LS (80.5%) by 0.9 percentage points. These results indicate that MaxSup is robust under class imbalance, especially in challenging settings with fewer samples in tail classes.
>
> To address the concern regarding the effectiveness of MaxSup on out-of-distribution (OOD) settings, we also conducted experiments on CIFAR10-C benchmark shown in Table R2 following settings in [2*]. Table R2 reports the performance of MaxSup and Label Smoothing (LS) on this benchmark using ResNet-50 as the backbone. Specifically, LS yields a better NLL (1.5730 vs. 1.8431), implying more confident probabilistic predictions. However, MaxSup achieves a better ECE (0.1479 vs. 0.1741), indicating better calibration of the predicted confidence scores. These results validate that **MaxSup remains effective on OOD datasets, achieving performance comparable to LS across all three metrics**.
>
> **Table R1**: Comparison of overall accuracy (%) (jointly considering many-shot, median-shot, and low-shot top-1 accuracy) for different loss strategies (Focal Loss vs Label Smoothing (LS) vs MaxSup) across datasets, imbalance levels, and backbones. Best performances are in bold.
>
> | **Dataset**           | **Split** | **Imbalance Ratio** | **Backbone** | **Focal Loss** | **+LS**  | **+MaxSup** |
> |-----------------------|-----------|----------------------|--------------|----------------|--------|------------|
> | Long-tailed CIFAR-10  | val       | 50                   | Resnet32     | 77.4           | 81.2   | **82.1**   |
> | Long-tailed CIFAR-10  | test      | 50                   | Resnet32     | 76.8           | 80.5   | **81.4**   |
> | Long-tailed CIFAR-10  | val       | 100                  | Resnet32     | 75.1           | 76.6   | **77.1**   |
> | Long-tailed CIFAR-10  | test      | 100                  | Resnet32     | 74.7           | 76.4   | **76.4**   |
>
>
> **Table R2**: Comparison of MaxSup and Label Smoothing (LS) on CIFAR10-C using Resnet50 as backbone. For all metrics, lower value is better, and best performances are in bold.
>
> | Metric         | MaxSup | LS |
> |----------------|--------|------------------|
> | Error (Corr)   | **0.3951** | **0.3951**           |
> | NLL (Corr)     | 1.8431 | **1.5730**           |
> | ECE (Corr)     | **0.1479** | 0.1741           |
>
> [1*] Tang, K., Huang, J., & Zhang, H. Long-tailed classification by keeping the good and removing the bad momentum causal effect. In NeurIPS 2020, pp. 1513–1524.
>
> [2*] Heinonen, M., Tran, B. H., Kampffmeyer, M., and Filippone, M. "Robust Classification by Coupling Data Mollification with Label Smoothing." AISTATS 2025.
>
> > **Q2**:  How about integrating the proposed method into other label smoothing methods? It would be synergized enough to increase the model performance
>
> Thank you for the suggestion. As shown in Table 14 in Appendix L (Ablation on the Weight Schedule), we combined the adaptive scheduler from Adaptive Label Smoothing [18] with our MaxSup method, which indeed led to consistent performance improvements. This indicates that there is potential synergy between MaxSup and existing label smoothing variants.
>
> We leave a more comprehensive exploration of such combinations—especially with more advanced or task-specific label smoothing techniques—as an interesting direction for future work.
>
> > **"Things to improve the paper that did not impact the score"**
>
> We are glad to resolve these formatting issues in the revised version.

---

> > ### Comment · Reviewer_cjw9 · 2025-08-05
> >
> > Thanks to the authors' efforts in conducting additional experiments about LT and OOD to address questions. The experimental results are interesting, but one concern still remains that the accuracy of minor classes may not have improved. (This concern should not impact the decision because the proposed method improves total accuracy in LT experiments.) However, this work has validated the effectiveness of the proposed method, and the authors have addressed other reviewers' concerns appropriately. For this reason, I decided to keep my rating

---

> > > ### Author Response · Authors · 2025-08-05
> > > **Author Follow-up Comment**
> > >
> > > **Dear Reviewer cjw9,**
> > >
> > > We sincerely thank you for acknowledging the effectiveness of our proposed method and for recognizing that the concerns have been appropriately addressed. We also appreciate your thoughtful observation regarding the performance on minor classes in the long-tailed (LT) experiments.
> > >
> > > To further support the discussion, we are happy to share the decoupled accuracy results (Many / Medium / Low) on the long-tailed CIFAR-10 dataset below:
> > >
> > > | **Dataset**           | **Split** | **Imbalance Ratio** | **Backbone** | **Method**        | **Overall** | **Many** | **Medium** | **Low** |
> > > |-----------------------|-----------|----------------------|--------------|-------------------|-------------|----------|------------|---------|
> > > | LT CIFAR-10           | val       | 50                   | ResNet32     | Focal Loss        | 77.4        | 76.0     | 89.7       | 0.0     |
> > > |                       |           |                      |              | Label Smoothing   | 81.2        | 81.6     | 77.0       | 0.0     |
> > > |                       |           |                      |              | **MaxSup**        | **82.1**    | 82.5    | 78.1       | 0.0     |
> > > | LT CIFAR-10           | test      | 50                   | ResNet32     | Focal Loss        | 76.8        | 75.3     | 90.4       | 0.0     |
> > > |                       |           |                      |              | Label Smoothing   | 80.5        | 81.1     | 75.4       | 0.0     |
> > > |                       |           |                      |              | **MaxSup**        | **81.4**    | 82.3     | 73.4       | 0.0     |
> > > | LT CIFAR-10           | val       | 100                  | ResNet32     | Focal Loss        | 75.1        | 71.8     | 88.3       | 0.0     |
> > > |                       |           |                      |              | Label Smoothing   | 76.6        | 80.6     | 60.7       | 0.0     |
> > > |                       |           |                      |              | **MaxSup**        | **77.1**    | 80.1     | 65.1       | 0.0     |
> > > | LT CIFAR-10           | test      | 100                  | ResNet32     | Focal Loss        | 74.7        | 71.6     | 87.2       | 0.0     |
> > > |                       |           |                      |              | Label Smoothing   | 76.4        | 80.8     | 59.0       | 0.0     |
> > > |                       |           |                      |              | **MaxSup**        | **76.4**    | 79.9     | 62.4       | 0.0     |
> > >
> > > In summary, MaxSup consistently improves performance on many-shot classes and achieves a better trade-off between many- and medium-shot accuracy, particularly under higher imbalance ratios. While it does not fully resolve the challenge of imbalanced classification—especially for minority classes—it shows positive effects and offers a promising direction for further extension.
> > >
> > > We hope these additional insights help clarify the behavior of MaxSup in long-tailed settings and contribute meaningfully to the scientific discussion around its merits.
> > >
> > > Thank you again for your constructive feedback and engagement throughout the review process.

---

### Decision · Program_Chairs · 2025-09-17

**Decision:**

Accept (oral)

**Comment:**

The paper received positive reviews, two accepts, and two borderline accepts. Many of the uncertainty calibration techniques use label smoothing and similar probability reshaping ideas. Hence, getting an understanding into the behaviour of such techniques and mitigation of shortcomings could be interesting for the community. AC notes that atleast one of the borderline accept is due to another paper in ICLR’25. AC had a brief look at the ICLR paper and found that though the two are focussed on a similar problem, the insights and mitigation seemed different.

Label smoothing, and the broader set of output probability reshaping, are techniques already getting used in variety of applications and hence any further insight into its functioning and failure points should be of wider interest to the community. The proposed technique looks a simple and elegant way of doing it, and hence the AC recommends accepting the paper. The AC is also happy to recommend the paper for oral presentation.